# Open-o3-Video: Grounded Video Reasoning with Explicit Spatio-Temporal Evidence

**Jiahao Meng** [1 2]   **Xiangtai Li** [2]   **Haochen Wang** [2 3]   **Yue Tan** [1]   **Tao Zhang** [2 4]   **Lingdong Kong** [2 5]   **Yunhai Tong** [1]   **Anran Wang** [2]   **Zhiyang Teng** [2]   **Yujing Wang** [2]   **Zhuochen Wang** [2]

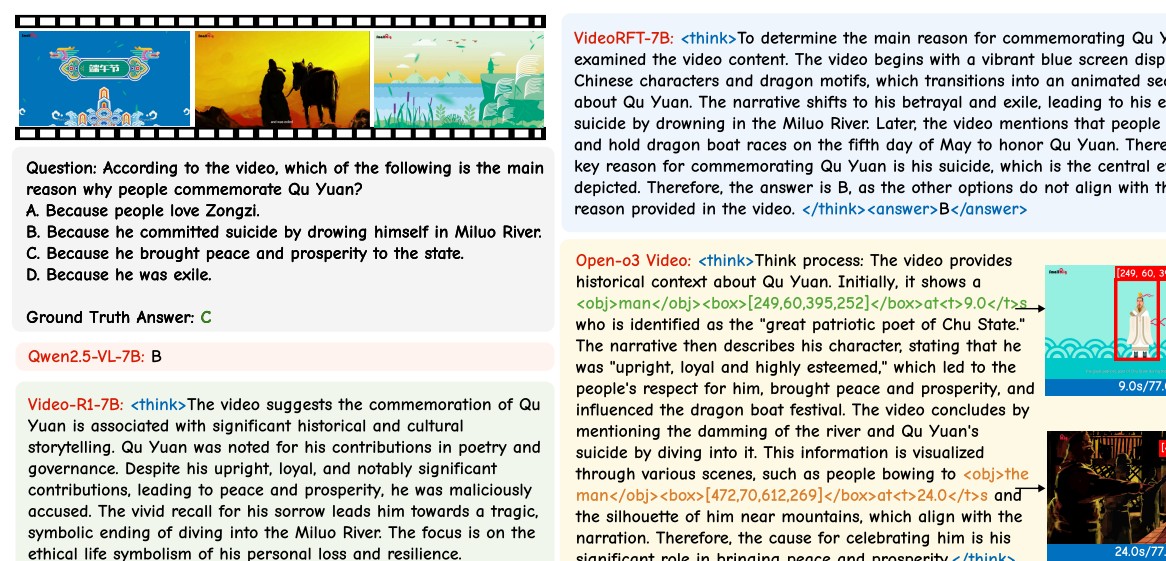

*Figure 1.* While prior video reasoning models (e.g., Video-R1 (Feng et al., 2025), VideoRFT (Wang et al., 2025c)) produce only textual rationales, **Open-o3-Video** natively integrates explicit spatio-temporal grounding into the reasoning process in a single forward pass, without relying on external tools. The model directly highlights supporting timestamps and object regions, yielding verifiable visual evidence for its predictions. Additional visualizations are provided in Appendix A.11.

## Abstract

Most video reasoning models only generate textual reasoning traces without indicating when and where key evidence appears. Recent models such as OpenAI-o3 have sparked wide interest in evidence-centered reasoning for images, yet extending this ability to videos is more challenging due to the need for joint temporal tracking and spatial localization across dynamic scenes. We introduce **Open-o3-Video**, a non-agent framework that integrates explicit spatio-temporal evidence into video reasoning by highlighting key timestamps, objects, and bounding boxes, making the reasoning process traceable and verifiable. To enable this capability, we first construct high-quality datasets **STGR** that provide unified spatio-temporal supervision, which is absent in existing resources. We further adopt a cold-start reinforcement learning strategy with specially designed rewards that jointly encourage answer accuracy, temporal alignment, and spatial precision. On the V-STAR benchmark, Open-o3-Video achieves state-of-the-art performance, improving mAM by 14.4% and mLGM by 24.2% over the Qwen2.5-VL baseline, and shows consistent gains across a range of video understanding benchmarks. Beyond accuracy, the grounded reasoning traces produced by Open-o3-Video support confidence-aware test-time scaling, improving answer reliability. The

[1]Peking University [2]ByteDance [3]Institute of Automation of the Chinese Academy of Sciences [4]Wuhan University [5]National University of Singapore. Correspondence to: Yunhai Tong <yh-tong@pku.edu.cn>.

code, model and datasets are publicly available at https://marinero4972.github.io/projects/Open-o3-Video/.

## 1. Introduction

Understanding complex video content is a long-standing goal for large multimodal models (Wang et al., 2025a; Team et al., 2025; Chen et al., 2025c; Zhang et al., 2024; Ye et al., 2025; Zhang et al., 2023; 2025d; Wang et al., 2025b), as videos encapsulate rich temporal dynamics and spatial interactions that far exceed the information in static images. While recent progress has improved performance on tasks such as video question answering (Bai et al., 2025b; Zhu et al., 2025; Zhang et al., 2025d; Li et al., 2024; Zhang et al., 2025a), building models that can perform reliable, fine-grained reasoning over long, cluttered scenes remains challenging.

Recent *"Thinking with Images"* attempts (OpenAI, 2025; Wang et al., 2026b;a; Zheng et al., 2026) leverage explicit operations (such as cropping, zoom-in, and region selection) to interleave detailed *visual evidence* with language, achieving superior performance on fine-grained image comprehension. This success motivates extending a similar paradigm to the video domain.

However, this extension is difficult and non-trivial due to the requirement for *coherent localization across both time and space* precisely. The complexity of dynamic scenes, *e.g.*, replete with motion, occlusions, and camera changes, makes it incredibly challenging to pinpoint when and where events of interest occur. As a result, previous attempts to incorporate explicit reasoning in video have often been limited to *textual rationales* (Feng et al., 2025; Wang et al., 2025c) or, coarse, *temporal-only* grounding (Li et al., 2025b; Wang et al., 2025d), failing to achieve the fine-grained spatio-temporal precision necessary for complex video reasoning. This gap is largely due to two interconnected obstacles: (1) the absence of *high-quality datasets* that provide joint spatio-temporal supervision for reasoning, and (2) the inherent difficulty of training a model to precisely localize objects in *time and space* simultaneously.

To address these challenges, we introduce **Open-o3-Video**, a framework that embeds *joint* spatio-temporal evidence directly into the reasoning process. Our first key contribution is the creation of a comprehensive training corpus designed to bridge this data gap. We have curated two training datasets, **STGR-CoT-30k** and **STGR-RL-36k**, for supervised fine-tuning and reinforcement learning, respectively. These datasets integrate existing temporal-only and spatial-only grounding resources *with 5.9k newly annotated high-quality spatio-temporal samples*. Each instance contains a question-answer pair, timestamped key frames, local-ized bounding boxes, and *a chain of thought that explicitly links the visual evidence to the reasoning steps*.

Building on this dataset, our second contribution is a two-stage training strategy with **adaptive temporal proximity** and **temporal gating** to stably and efficiently optimize the model's spatio-temporal reasoning capability. Although the model has acquired preliminary capabilities for generating structured, grounded chains of thought during the supervised fine-tuning stage, the subsequent reinforcement learning stage still cannot achieve stable training due to a critical *spatial collapse* issue. This is because spatial grounding rewards are usually conditioned on correctly identifying the timestamp. When temporal predictions are imprecise in the early stages, this leads to *near-zero spatial rewards*, stalling the learning process for localization ability. Therefore, we propose a novel *adaptive temporal proximity* technique that relaxes the temporal requirement during early training to reduce reward sparsity and gradually increases the precision demand over time. This training strategy prevents premature saturation of the temporal reward signal and ensures that predicted timestamps continue to approach the ground truth, which is crucial for reliable spatial evaluation. In parallel, a complementary *temporal gating* mechanism computes spatial rewards only when temporal predictions are sufficiently accurate, preventing irrelevant objects from being rewarded and enforcing precise spatio-temporal alignment. Together, these mechanisms provide dense yet reliable feedback, forming a smoother learning curriculum that progressively strengthens both temporal accuracy and spatial grounding.

Through this combination of curated data and our training procedure, as shown in Figure 1, Open-o3-Video produces reasoning that is accurate, interpretable, and grounded in the visual evidence. We evaluate Open-o3-Video on the V-STAR benchmark (Cheng et al., 2026) and other video understanding tasks. On **V-STAR**, our model achieves state-of-the-art performance, surpassing GPT-4o and improving over Qwen2.5-VL by **+14.4%** mAM and **+24.2%** mLGM with a small amount of training data. Beyond V-STAR, Open-o3-Video also delivers consistent gains on VideoMME, World-Sense, VideoMMMU, LongVideo-Reason-eval, and TVG-Bench, demonstrating advantages in long-video reasoning, perception-oriented tasks, and fine-grained temporal localization. We further find that the generated evidence is informative, as removing evidence-aligned frames causes larger performance degradation, and it can also be leveraged for test-time scaling, where evidence-based confidence-aware voting outperforms majority voting (e.g., +1.2% on World-Sense and 1.0% on VideoMMMU).

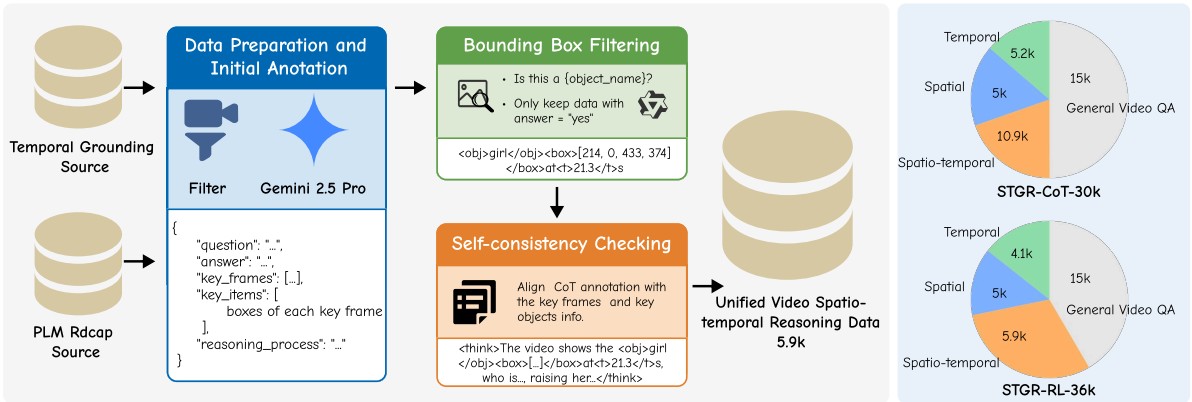

*Figure 2.* Overview of our data construction pipeline and dataset composition. **Left**: The annotation pipeline includes Gemini 2.5 Pro initial annotation, bounding box filtering, and self-consistency checking. **Right**: Distribution of data categories in STGR-CoT-30k (SFT) and STGR-RL-36k (RL), showing a balanced coverage across temporal, spatial, spatio-temporal, and general QA.

## 2. Related works

**Video Reasoning.** Recent advances in video reasoning (Feng et al., 2025; Li et al., 2025b; Wang et al., 2025e;c; Zhang et al., 2025c; Xie et al., 2025; Chen et al., 2025b; Zhang et al., 2026; Park et al., 2025; Dang et al., 2025) have largely been driven by reinforcement learning based post-training on multi-modal large language models (MLLMs), which encourages models to move beyond direct question answering and exhibit step-by-step reasoning. Video-R1 (Feng et al., 2025) shows that temporal-aware GRPO with curated reasoning data improves video understanding benchmarks, while VideoChat-R1 (Li et al., 2025b) extends to spatio-temporal perception tasks such as grounding and tracking without harming QA. Other variants, including Video-RTS (Wang et al., 2025e) and DeepVideo-R1 (Park et al., 2025), combine reinforcement learning with test-time scaling or difficulty-aware regularization to better exploit temporal information. Despite their success, most methods rely on text-only reasoning. However, our approach incorporates spatio-temporal evidence for transparent and verifiable grounding.

**Temporal and Spatial Grounding in Video.** The problem of locating when and where relevant evidence appears in a video has attracted increasing attention, leading to progress in both temporal and spatial grounding (Wang et al., 2025d; 2026c; Chen et al., 2025a; Guo et al., 2025; Ouyang et al., 2025; Li et al., 2025c;b). Temporal grounding methods such as Time-R1 (Wang et al., 2025d) and TVG-R1 (Chen et al., 2025a) leverage verifiable rewards and curated RL data to improve temporal localization, while spatial grounding approaches like SpaceR (Ouyang et al., 2025) focus on object-centric localization and geometric reasoning. Several works further explore spatio-temporal localization through architectural or training designs, including STCAT (Jin et al., 2022), LRR (Bhattacharyya et al., 2024), EgoMask (Liang et al., 2025), and LLaVA-ST (Li et al., 2025a). However, existing methods do not align spatio-temporal localization with chain-of-thought reasoning. Our approach addresses this gap by explicitly linking object regions with temporal positions and incorporating spatio-temporal evidence into reasoning, enabling more verifiable video understanding.

**Thinking with Images.** A growing line of research (OpenAI, 2025; Zheng et al., 2026; Wang et al., 2026a;b; Fan et al., 2025) explores how multi-modal models improve reasoning by performing explicit visual operations such as cropping, zoom-in, and region selection, thereby producing intermediate evidence that is consumed within the reasoning chain. OpenAI-o3 (OpenAI, 2025) formalizes "thinking with images," while DeepEyes (Zheng et al., 2026) shows end-to-end RL can incentivize image–tool reasoning, and TreeBench (Wang et al., 2026a) provides methodology for traceable, box-level evidence. These advances demonstrate the promise of evidence-centric visual reasoning but are largely image-centric. Extending to videos adds challenges in temporal consistency, motion, and fine-grained event alignment. Several concurrent works discussed in Appendix A.2 like VITAL (Zhang et al., 2026) and Conan (Ouyang et al., 2026) adapt the paradigm via an agent-based, tool-augmented RL pipeline, yielding gains but relying on external orchestration. In contrast, our framework "thinks with frames" in a single round of inference, directly emitting timestamped crops and bounding boxes as evidence without complex tool pipelines.

## 3. STGR Data Construction

### 3.1. Data Source and Statistics

Building robust spatio-temporal reasoning models requires training signals that jointly supervise *when* and *where* evidence appears and how it is used in reasoning. Existing

resources fall short in three ways: (i) temporal-only grounding datasets provide time spans but lack object regions; (ii) spatial or frame-level caption corpora offer boxes on isolated frames without timestamps; and (iii) most lack a chain of thought that *explicitly* ties objects and timestamps to the answer. These gaps make it impossible to learn coherent localization in dynamic scenes and to compute verifiable rewards for RL, since temporal and spatial supervision are not synchronized and reasoning traces are text-only.

To bridge this gap, we curate two complementary corpora: **STGR-CoT-30k** for supervised fine-tuning (SFT) and **STGR-RL-36k** for reinforcement learning (RL). Both combine existing temporal-only and spatial-only resources *with 5.9k newly annotated, high-quality spatio-temporal samples* produced by our pipeline (Sec. 3.2). Each new instance includes a question–answer pair, timestamped key frames, localized boxes, and a structured chain of thought that links visual evidence to reasoning steps. This design provides synchronized temporal and spatial supervision for SFT to acquire grounded reasoning formats and reliable, verifiable signals for RL to optimize alignment under complex video dynamics.

The SFT corpus consists of four components: (i) 4.1k temporal grounding CoT samples (TVG-Coldstart) (Chen et al., 2025a), (ii) 5k spatial grounding CoT samples (TreeVGR-SFT) (Wang et al., 2026a), (iii) 5.9k spatio-temporal samples curated by us, including 3.9k from temporal grounding datasets (video sources are shown in Appendix A.3) and 2k from PLM-Rdcap (Cho et al., 2025), and (iv) 15k Video-R1-CoT samples (Feng et al., 2025). The RL corpus further expands diversity: (i) 5.2k temporal grounding samples, including 2.3k from Time-R1 (Wang et al., 2025d) and 2.9k from TVG-RL (Chen et al., 2025a), (ii) 5k spatial grounding samples from VisCoT (Shao et al., 2024a), (iii) 10.9k spatio-temporal samples, comprising our 5.9k constructed data (via the pipeline) and an additional 5k filtered from VideoEspresso (Han et al., 2025) with consistency checks, and (iv) 15k Video-R1 samples (Feng et al., 2025).

Overall, as shown in Figure 2 (right), the SFT set covers 13.7% temporal, 16.7% spatial, 19.7% spatio-temporal, and 50.0% general QA data, while the RL set includes 14.4% temporal, 13.9% spatial, 30.3% spatio-temporal, and 41.7% QA data. This design ensures that both phases expose the model to diverse supervisory signals while emphasizing spatio-temporal reasoning as the central capability. More details of training data are provided in the Appendix A.3.

### 3.2. Data Annotation Pipeline

Spatio-temporal reasoning requires chain-of-thought data that include both temporal spans and spatial grounding. We construct 5.9k such samples by combining temporal grounding datasets with PLM-Rdcap sources (Figure 2, left). The

pipeline follows three stages below.

**Data Preparation and Initial Annotation.** We begin by collecting two types of sources: temporal grounding datasets and PLM-Rdcap data that provide region-level dense captions. All videos are passed through the Gemini 2.5 Pro (Comanici et al., 2025) API, with carefully designed prompts (shown in Appendix A.4) to generate structured annotations. Each annotation contains (i) a question-answer pair centered on a specific object or person, (ii) one to five key frames sampled from the annotated segment, (iii) bounding boxes for one to three salient objects in each key frame, and (iv) a reasoning process that must reference every object with explicit format: `<obj>object_name</obj><box>[x_min, y_min, x_max, y_max]</box>at<t>timestamp</t>s`.

**Bounding Box Filtering.** Initial annotations may contain noisy or incorrect boxes. We filter them with two rules: (i) boxes covering over 80% of the frame are removed as uninformative; (ii) each crop is verified by Qwen2.5-VL-7B (Bai et al., 2025b) with the query "Is this a {object_name}?". Only samples that answered "yes" are kept, ensuring that object mentions match the validated boxes.

**Self-consistency Checking and Quality Control.** Our consistency checking enforces alignment between timestamps, bounding boxes, and the spatio-temporal reasoning chain. For each annotated sample, we verify that all temporal and spatial references appearing in the reasoning text are covered by the corresponding "key_frames" and "key_items" annotations. Samples with missing elements are discarded. We further assess the relevance between each reasoning sentence and its associated visual evidence by cropping the referenced region and querying Qwen2.5-VL to judge semantic consistency. Samples with inconsistent visual-textual alignment are removed. These checks improve annotation quality and provide reliable supervision for cold-start grounded training.

## 4. Open-o3-Video

As shown in Figure 3, our training recipe comprises two stages: a cold-start initialization phase followed by reinforcement learning that enhances spatio-temporal reasoning through carefully designed rewards with **adaptive temporal proximity and temporal gating** mechanisms.

### 4.1. Cold Start Initialization

We initialize our model from Qwen2.5-VL-7B (Bai et al., 2025b), and fine-tune it on the constructed STGR-CoT-30k corpus. This cold-start stage equips the model with basic spatio-temporal grounding and structured reasoning capabilities, reducing reward sparsity and stabilizing subsequent reinforcement learning.

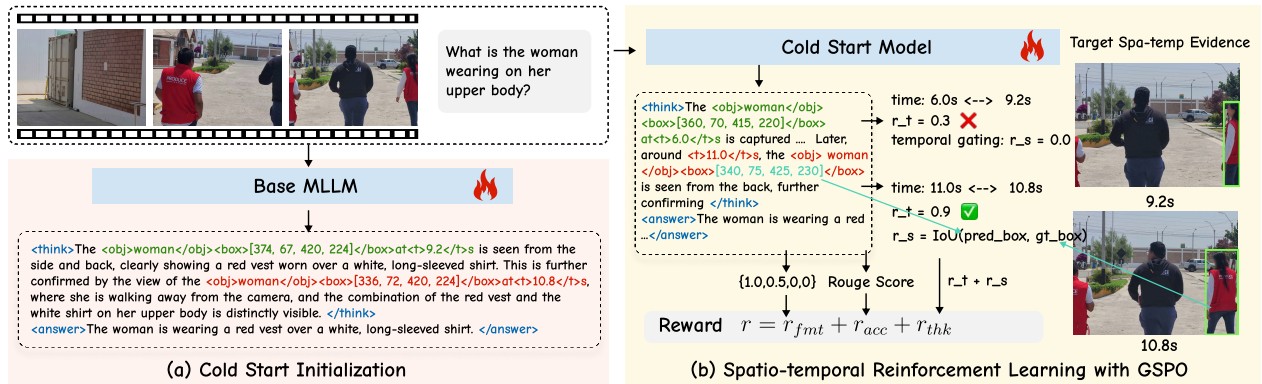

*Figure 3.* Overview of Open-o3-Video. We adopt a two-stage training paradigm: (a) cold-start initialization to learn structured, grounded outputs; (b) reinforcement learning with a composite reward that sharpens temporal alignment and spatial precision with adaptive temporal proximity and temporal gating.

## 4.2. Reinforcement Learning with GSPO

We adopt Group Sequence Policy Optimization (GSPO) (Zheng et al., 2025) as our reinforcement learning algorithm. Compared with GRPO (Shao et al., 2024b), which operates at the token level, GSPO defines importance ratios and clipping at the sequence level, ensuring that optimization is aligned with sequence-level rewards. This eliminates high-variance token-wise corrections, stabilizes long-horizon training, and avoids collapse in chain-of-thought reasoning. In our video spatio-temporal grounded reasoning setting, rewards are defined over complete reasoning traces that include timestamps and bounding boxes, rather than individual tokens. GSPO better matches by optimizing whole sequences as atomic units, allowing the policy to consistently improve global grounding quality instead of overfitting to local token-level signals. As a result, GSPO yields higher grounding accuracy and more stable training than GRPO in our experiments (Section 5.2).

During training, given a video-question pair $x$, each generated response $y$ is evaluated with a scalar reward $r(x, y)$ that reflects both correctness and reasoning quality. This reward serves as the optimization signal in GSPO, and more details of the GSPO algorithm are provided in Appendix A.9.

## 4.3. Reward Design

For each query–completion pair $(x, y)$, the scalar reward is defined as

$$r(x, y) = r_{\text{acc}}(x, y) + r_{\text{thk}}(x, y) + r_{\text{fmt}}(x, y), \quad (1)$$

which is group-normalized to obtain the advantage used by GSPO. Below, we describe the three components.

**Accuracy Reward $r_{\text{acc}}$.** Since the training data span multiple tasks, we design task-specific accuracy rewards. Let

$\tau \in \{\text{MCQ}, \text{QA}, \text{SG}, \text{TG}\}$ denote the task type, where MCQ is a multiple-choice question, QA is free-form question answering, SG is spatial grounding, and TG is temporal grounding. For spatial grounding, we denote the predicted and ground-truth bounding boxes by $B^{\text{pred}}$ and $B^{\text{gt}}$, respectively. For temporal grounding, we denote the predicted and ground-truth temporal segments by $S^{\text{pred}} = [s^{\text{pred}}, e^{\text{pred}}]$ and $S^{\text{gt}} = [s^{\text{gt}}, e^{\text{gt}}]$. We then define:

$$r_{\text{acc}}(x, y) = \begin{cases} \mathbb{I}\big[y^{\text{pred}} = y^{\text{gt}}\big], & \tau = \text{MCQ}, \\ \text{ROUGE}\big(y^{\text{pred}}, y^{\text{gt}}\big), & \tau = \text{QA}, \\ \text{vIoU}\big(B^{\text{pred}}, B^{\text{gt}}\big), & \tau = \text{SG}, \\ \text{tIoU}\big(S^{\text{pred}}, S^{\text{gt}}\big), & \tau = \text{TG}. \end{cases} \quad (2)$$

**Thinking Reward $r_{\text{thk}}$.** We define the thinking reward as the sum of temporal and spatial terms:

$$r_{\text{thk}}(x, y) = r_{\text{t}}(x, y) + r_{\text{s}}(x, y). \quad (3)$$

*Temporal term with adaptive temporal proximity.* Let $M$ be the number of timestamps $\{t_m\}_{m=1}^M$ parsed from <think>. The temporal reward $r_{\text{t}}(x, y)$ depends on the supervision type $\tau_t \in \{\text{Int}, \text{Pt}, \varnothing\}$: Interval supervision (Int) provides a ground-truth span $[s^{\text{gt}}, e^{\text{gt}}]$, point supervision (Pt) provides ground-truth timestamps $\{t_j^{\text{gt}}\}$; and $\varnothing$ indicates no timestamp evidence. For point supervision, we define the closest temporal distance $\Delta t_m = \min_j |t_m - t_j^{\text{gt}}|$.

$$r_{\text{t}}(x, y) = \begin{cases} \dfrac{1}{M} \sum_{m=1}^M \mathbf{1}\big[s^{\text{gt}} \leq t_m \leq e^{\text{gt}}\big], & \tau_t = \text{Int}, \\ \dfrac{1}{M} \sum_{m=1}^M \exp\bigg(-\dfrac{\Delta t_m^2}{2\sigma^2}\bigg), & \tau_t = \text{Pt}, \\ 0, & \tau_t = \varnothing. \end{cases} \quad (4)$$

A key difficulty is that spatial rewards depend on accurate temporal predictions: IoU can only be computed reliably

when the timestamp is close to the ground truth. If the temporal constraint is too strict (i.e., $\sigma$ very small), the model receives little reward when its early temporal predictions are inaccurate, which slows down temporal learning and, in turn, prevents spatial grounding from being learned effectively. Conversely, if the constraint is always loose (i.e., $\sigma$ large), temporal rewards quickly saturate and stop driving predicted timestamps closer to the ground truth, which again undermines spatial reward reliability. To resolve this trade-off, we propose **adaptive temporal proximity**: $\sigma$ is large in early training to provide dense signals, and gradually decreases to enforce stricter alignment. This strategy ensures that the model first obtains stable gradients and later achieves precise timestamping, providing a solid foundation for spatial evaluation.

*Spatial term with temporal gating.* For each predicted timestamp $t_m$, let $j^\star(m) = \arg\min_j |t_m - t_j^{\text{gt}}|$ be the nearest annotated time. Let $\mathcal{B}_m$ be the predicted boxes at $t_m$ and $\mathcal{B}_{j^\star(m)}^{\text{gt}}$ be the ground-truth boxes at the matched frame. We define the per-frame maximal overlap as $v_m = \max_{b \in \mathcal{B}_m,\, b^{\text{gt}} \in \mathcal{B}_{j^\star(m)}^{\text{gt}}} \text{IoU}(b, b^{\text{gt}})$. The spatial reward is

$$r_{\text{s}}(x,y) = \frac{1}{M} \sum_{m=1}^{M} \mathbf{1}\Big[ |t_m - t_{j^\star(m)}^{\text{gt}}| \leq \tau \Big] \cdot v_m, \quad (5)$$

where $\tau$ is a temporal threshold. We propose a **temporal gating** mechanism to guarantee the reliability of spatial supervision. Specifically, spatial rewards are only computed when temporal predictions are sufficiently close to the ground truth. This prevents rewarding salient but irrelevant objects at wrong timestamps, enforces spatio-temporal alignment, and ultimately improves both the interpretability and reliability of the reasoning process. Together, adaptive temporal proximity and temporal gating provide complementary solutions: the former provides stable, progressive temporal supervision, while the latter ensures accurate, trustworthy spatial evaluation.

**Format Reward** $r_{\text{fmt}}$. Strict usage of `<think>` and `<answer>` with correct `<obj>` `<box>` `<t>` gives 1.0. Having only `<think>` and `<answer>` yields 0.5. Otherwise, the reward is 0.0.

# 5. Experiments

**Implementation Details.** We build upon the **Qwen2.5-VL-7B** model and train on 8 NVIDIA H100 GPUs. During training, we uniformly sample 16 frames from each video, where each frame has a resolution not exceeding $128 \times 28 \times 28$. If annotated key frames are available, they are inserted in addition to the uniformly sampled frames. To strengthen the model's perception of temporal information, we prepend each frame with its absolute timestamp. More implementation details are provided in Appendix A.1.

**Benchmarks.** We adopt V-STAR (Cheng et al., 2026) as the main benchmark, since it is specifically designed to measure spatio-temporal grounding in videos. Unlike conventional video QA datasets, V-STAR requires models not only to answer questions but also localize *when* and *where* the supporting evidence occurs. It introduces two structured reasoning chains ("what–when–where" and "what–where–when") and composite metrics that combine accuracy with temporal and spatial IoU, thereby enabling comprehensive evaluation of spatio-temporal reasoning. We further evaluate on broader video understanding benchmarks. VideoMME (Fu et al., 2025) and VideoMMMU (Hu et al., 2025) assess general video QA and multimodal comprehension across diverse domains, while WorldSense (Hong et al., 2025) emphasizes integrating multimodal signals with commonsense reasoning, and LongVideo-Reason-eval (Chen et al., 2025b) evaluates long-range reasoning on videos. In addition, TVG-Bench (Wang et al., 2025d) focuses on fine-grained temporal localization, STAR (Wu et al., 2024) tests situated reasoning, and CameraBench (Lin et al., 2025) measures robustness under diverse camera motions.

## 5.1. Main Results

**Results on V-STAR.** On the V-STAR benchmark, we compare our method with three groups of baselines: (i) closed-source commercial models such as GPT-4o (OpenAI, 2024), Gemini-2-Flash (Team et al., 2024) and Gemini-3-Pro (Google, 2025), which represent the current frontier of proprietary video LLMs. (ii) open-source general-purpose video understanding models, including Video-LLAMA3 (Zhang et al., 2025a), LLaVA-Video (Zhang et al., 2025d), VideoChat2 (Li et al., 2024), Oryx-1.5-7B (Liu et al., 2025), InternVL-2.5-8B (Chen et al., 2024), and Qwen2.5-VL-7B (Bai et al., 2025b). (iii) task-specialized approaches such as TRACE (Guo et al., 2025), designed for temporal video grounding, and Sa2VA (Yuan et al., 2025), optimized for fine-grained spatial grounding. As summarized in Table 1, our model consistently outperforms the baseline across different evaluation dimensions. In video question answering (*What*), our model achieves an accuracy of 61.03, representing a +27.6% point improvement over Qwen2.5-VL-7B. For temporal grounding (*When*), we report strong gains on both reasoning chains: Chain1 (*what–when–where*) improves by +9.1% points and Chain2 (*what–where–when*) by +10.2% points, showing robust performance regardless of the reasoning order. For spatial grounding (*Where*), our method surpasses the baseline by +8.4% points on Chain1 and +3.5% points on Chain2. Overall, compared with the Qwen2.5-VL baseline, our model improves performance by +14.4% mAM and +24.2% mLGM on V-STAR. It further surpasses proprietary models such as GPT-4o (OpenAI, 2024), Gemini-2-Flash (Comanici et al., 2025) and Gemini-3-Pro (Google, 2025), achieving state-of-

*Table 1.* Performance on the **V-STAR** benchmark, which evaluates **spatio-temporal** reasoning across three dimensions. Chain1 denotes *what–when–where*, while Chain2 corresponds to *what–where–when*. mAM is the average of the arithmetic mean, and mLGM is the average of the modified logarithmic geometric mean, combining temporal and spatial alignment. $^{*}$ indicates we re-evaluate using the vLLM framework with 16 sampled frames. Bold numbers denote the best results, while underlined numbers indicate the second best.

| Model | What | When (Temporal IoU) | | Where (Visual IoU) | | Overall | |
|---|---|---|---|---|---|---|---|
| | Acc | Chain1 | Chain2 | Chain1 | Chain2 | mAM | mLGM |
| GPT-4o | 60.8 | 16.7 | 12.8 | 6.5 | 3.0 | 26.8 | 38.2 |
| Gemini-2-Flash | 53.0 | **24.5** | 23.8 | 4.6 | 2.2 | 26.9 | 35.6 |
| Gemini-3-Pro | 59.1 | 24.2 | 23.8 | 7.2 | 4.8 | 29.9 | 41.2 |
| Video-LLaMA3 | 41.9 | 23.0 | 23.1 | 0.9 | 0.2 | 21.7 | 27.0 |
| LLaVA-Video | 49.5 | 10.5 | 12.2 | 1.9 | 1.3 | 20.8 | 27.3 |
| VideoChat2 | 36.2 | 13.7 | 12.5 | 2.5 | 1.0 | 17.0 | 20.3 |
| Oryx-1.5-7B | 20.5 | 13.5 | 14.8 | 10.1 | 3.5 | 15.1 | 13.8 |
| InternVL-2.5-8B | 44.2 | 8.7 | 7.8 | 0.7 | 0.1 | 17.6 | 24.9 |
| Qwen2.5-VL-7B$^{*}$(base) | 33.5 | 15.4 | 13.8 | 17.0 | 2.5 | 19.3 | 22.4 |
| TRACE | 17.6 | 19.1 | 17.1 | 0.0 | 0.0 | 12.0 | 13.3 |
| Sa2VA-8B | 16.4 | 0.1 | 0.0 | **32.3** | **37.5** | 17.1 | 20.3 |
| Open-o3-Video-7B (Ours) | **61.0** | **24.5** | **24.0** | 25.4 | 6.0 | **33.7** | **46.6** |
| Δ *vs.* Qwen2.5-VL-7B | ↑ 27.5 | ↑ 9.1 | ↑ 10.2 | ↑ 8.4 | ↑ 3.5 | ↑ 14.4 | ↑ 24.2 |

*Table 2.* Performance across different video understanding, reasoning, and temporal grounding benchmarks. "LRR" refers to LongVideo-Reason-eval Benchmark. Open-o3-Video achieves comparable or even superior results to other video understanding and reasoning models, while providing more intuitive spatiotemporal evidence. Evaluation results on more benchmarks are provided in Appendix A.7.

| Model | VideoMME | | WorldSense | | VideoMMMU | | LRR | TVGBench | Avg |
|---|---|---|---|---|---|---|---|---|---|
| | Overall | Long | Overall | Recognition | Overall | Perception | Acc | mIoU | |
| GPT-4o | 71.9 | - | 42.6 | - | 61.2 | 66.0 | - | - | - |
| VideoLLaMA3-7B | 60.6 | 48.7 | 37.3 | 38.1 | 46.5 | 59.7 | 59.8 | **22.2** | 45.3 |
| InternVL-2.5-8B | 62.3 | 51.2 | **39.6** | **38.5** | 42.4 | 57.0 | 62.0 | 6.3 | 42.5 |
| Qwen2.5-VL-7B (Base) | 62.4 | 50.8 | 36.1 | 33.7 | 51.2 | 64.7 | 59.3 | 16.3 | 45.1 |
| VideoRFT-7B | 59.8 | 50.7 | 38.2 | 36.6 | 51.1 | 66.0 | 69.4 | 14.3 | 46.6 |
| VideoR1-7B | 61.4 | 50.6 | 35.5 | 32.8 | **52.4** | 65.3 | 68.9 | 9.6 | 45.6 |
| Open-o3-Video-7B (Ours) | **63.6** | **54.9** | 37.5 | 36.8 | 52.3 | **68.0** | 69.4 | 20.8 | **48.7** |
| Δ *vs.* Qwen2.5-VL-7B | ↑ 1.2 | ↑ 4.1 | ↑ 1.4 | ↑ 3.1 | ↑ 1.1 | ↑ 3.3 | ↑ 10.1 | ↑ 4.5 | ↑ 3.6 |

the-art performance. By extracting key frames and precise bounding boxes, Open-o3-Video brings o3-style, evidence-guided reasoning to videos, supplying more reliable and verifiable visual evidence during inference. We further train our framework on Qwen3-VL (Bai et al., 2025a) models and observe consistent gains on V-STAR. Detailed results are provided in Appendix A.6.

**Results on General Video Understanding and Temporal Grounding Benchmarks.** We further evaluate our method on a broad suite of video understanding benchmarks, comparing against three categories of baselines: (i) closed-source models such as GPT-4o (OpenAI, 2024), (ii) open-source general video LLMs including Qwen2.5-VL-7B (Bai et al., 2025b), Video-LLAMA3 (Zhang et al., 2025a), and InternVL-2.5-8B (Chen et al., 2024), and (iii) recent reasoning-focused models such as VideoRFT-7B (Wang et al., 2025c) and VideoR1-7B (Feng et al., 2025),

which treat video understanding as text-only reasoning. In contrast, our method combines reasoning with explicit spatio-temporal grounding, enabling evidence-based inference. As shown in Table 2, Open-o3-Video achieves consistent improvements across all datasets. Across VideoMME, WorldSense, and VideoMMMU, our model shows consistent gains over Qwen2.5-VL-7B, with notable improvements on long videos (+4.1%) and perception-related tasks (+3.1% on WorldSense recognition and +3.3% on VideoMMMU perception), highlighting enhanced temporal reasoning and perceptual grounding. For long-range video reasoning, our model achieves 69.4% accuracy on the LongVideoReason-eval benchmark (LRR), and outperforms the baseline by +10.1%. Compared with dedicated video reasoning methods, our model achieves results comparable to or even superior to theirs while providing more interpretable evidence in its reasoning process. On TVGBench, which directly measures

*Table 3.* Ablation on Different training strategies.

| Setting | mAM | mLGM |
|---|---|---|
| Baseline (Qwen2.5-VL-7B) | 19.3 | 22.4 |
| Pure SFT | 28.5 | 37.1 |
| Pure RL (GSPO) | 30.4 | 40.7 |
| SFT+RL (GRPO) | 32.8 | 45.3 |
| SFT+RL (GSPO) (Ours) | **33.7** | **46.6** |

*Table 4.* Impact of two reward designs in the thinking reward.

| Ada. | Gat. | mAM | mLGM |
|---|---|---|---|
| $\times$ | $\checkmark$ | 33.0 | 45.2 |
| $\checkmark$ | $\times$ | 32.3 | 44.9 |
| $\checkmark$ | $\checkmark$ | **33.7** | **46.6** |

temporal grounding, our model surpasses the baseline by a large margin (+4.5% mIoU), indicating gains in temporal localization. These results show that our approach **maintains the QA strength of general video LLMs** while enhancing the spatio-temporal grounding capability.

### 5.2. Ablation and Analysis

**Training strategy: RL provides larger gains than SFT, while their combination yields the best results, with GSPO offering the most stable improvements.** As shown in Table 3, on the V-STAR benchmark, both SFT and RL substantially improve grounding over the base model. RL outperforms SFT (+2.1% mAM, +4.6% mLGM) by directly optimizing temporal and spatial alignment, while SFT ensures stable reasoning formats and basic grounding under supervision. Their combination is highly synergistic, reaching 33.7% mAM and 46.6% mLGM. Within this joint training, GSPO further surpasses GRPO (+0.9% mAM, +1.3% mLGM) by providing more stable rewards and better long-horizon temporal localization (+2.9% Chain1 tIoU). More ablations about training are provided in Appendix A.5.

**Reward design: Both adaptive temporal proximity and temporal gating are effective.** In the thinking reward, we introduce two mechanisms: adaptive temporal proximity (**Ada.**) and temporal gating (**Gat.**). To validate their effectiveness, we conduct ablation experiments on the V-STAR benchmark, shown in Table 4. Removing the proximity reward reduces performance by 0.7% mAM and 1.4% mLGM, showing that adaptive scaling helps the model better align predicted timestamps with annotated windows. Removing temporal gating causes larger drops of 1.4% mAM and 1.7% mLGM, confirming that gating is crucial for filtering irrelevant segments and preventing noisy spatial boxes. These results verify that our reward design effectively couples temporal and spatial grounding, leading to strong performance.

**Training data: High-quality spatio-temporal annotations boost grounding.** Without spatio-temporal supervision, the model exhibits substantially weaker performance,

*Table 5.* Impact of different spatio-temporal grounded reasoning data for training data. More ablations about training data are provided in the Appendix A.3.

| Training data | mAM | mLGM |
|---|---|---|
| w/o spatio-temporal data | 28.3 | 36.2 |
| + VideoEspresso | 31.1 | 43.6 |
| + Our annotated data | **33.7** | **46.6** |

*Table 6.* Effectiveness of the extracted temporal evidence.

| Setting | Accuracy (%) |
|---|---|
| Uniform sampling (64 frames) | **68.7** |
| Removed predicted evidence | 66.0 (-2.7) |
| Random removal | 68.0 (-0.7) |

underscoring the necessity of Spatio-temporal annotations for effective grounding. As shown in Table 5, incorporating 9.6k filtered and rewritten *VideoEspresso* (Han et al., 2025) samples enables the model to perform basic spatio-temporal reasoning, leading to improvements of +2.8% mAM and +7.4% mLGM on V-STAR. Moreover, further adding our spatio-temporal annotations can yield gains of +5.4% mAM and +10.4% mLGM. This shows the effectiveness of our annotation pipeline and the critical role of high-quality spatio-temporal supervision.

**Evidence faithfulness: Generated spatio-temporal evidence is concise and informative.** We analyze the generated evidence on a randomly sampled VideoMME subset of 150 questions. On average, the model produces 1.15 bounding boxes and 1.37 timestamps per instance. To assess the relevance of the identified evidence, we remove two frames temporally closest to each predicted timestamp. As shown in Table 6, removing evidence-related frames causes a larger performance drop than randomly removing the same number of frames, indicating that the generated evidence can capture some informative visual signals.

**Test-time scaling with grounded evidence: Confidence-aware voting with Open-o3-Video outperforms naive majority voting.** Inspired by the scoring and adaptive voting mechanisms for video reasoning in CyberV (Meng et al., 2025), we introduce a confidence-aware voting scheme that leverages grounded evidence to verify predictions at inference, as shown in Figure 4 in the appendix. Details, including scoring schemes, prompts, and results on WorldSense and VideoMMMU, are provided in Appendix A.10.

### 6. Conclusion

We introduce **Open-o3-Video**, a grounded video reasoning framework that enables a single model to generate explicit spatio-temporal evidence (timestamps and bounding boxes) as part of its reasoning process, without relying on external models or tools. To the best of our knowledge, we are the first to achieve this ability in Video MLLMs.

With carefully curated high-quality training data, a two-stage strategy combining supervised fine-tuning and GSPO-based reinforcement learning, and novel thinking rewards incorporating adaptive temporal proximity and temporal gating, our method substantially improves answer accuracy, temporal alignment, and spatial grounding. Comprehensive experiments demonstrate that Open-o3-Video achieves state-of-the-art performance on the V-STAR benchmark, surpassing strong baselines including Gemini-3-Pro, while remaining broadly competitive across diverse video understanding tasks.

**Limitations and Future Works.** Despite these results, our approach still has limitations in longer videos, complex scenes, and small-object grounding, where high-quality spatio-temporal supervision remains scarce. It also does not yet use audio, speech, or broader contextual evidence, which can be important for video reasoning. Future work will extend the evidence space and improve reasoning under more complex video conditions. We provide a fuller discussion in Appendix A.12.

## Acknowledgement

This work is supported by the National Key Research and Development Program of China (No. 2023YFC3807600).

## Impact Statement

We introduce a framework for grounded video reasoning that links model predictions to explicit spatio-temporal visual evidence. This o3-like paradigm improves the interpretability and reliability of video reasoning models. However, explicit spatio-temporal evidence may also be misused for privacy-sensitive localization, targeted video manipulation, or more realistic deepfakes when combined with harmful generation systems. We therefore release and apply this work only for research use, follow dataset licenses, and avoid private or personally identifiable data. Future deployment should include access control, misuse monitoring, and clear user guidance for responsible use.

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

# A. Appendix

**Overview.** This appendix provides additional details and analyses to complement the main paper.

## A.1. More Implementation Details

The training process of Open-o3-Video consists of two stages. In the cold-start stage, we train on the STGR-CoT-30k dataset for one epoch with a learning rate of $1 \times 10^{-6}$. In the GSPO stage, we further train on the STGR-RL-36k dataset for one epoch, also with a learning rate of $1 \times 10^{-6}$. For the thinking reward, the standard deviation parameter $\sigma$ is annealed from 4 to 1 and then kept constant. The gating mechanism employs a temporal threshold $\tau$ of 3s. At test time, we employ the vLLM framework, requiring the model to first produce a spatio-temporal grounded reasoning process, followed by the final answer. For evaluation on V-STAR, we uniformly sample 16 frames per video, and for other video understanding benchmarks, we uniformly sample 64 frames. Additional comparisons and analyses on inference frame rates are provided in Appendix A.8.

## A.2. More Related Works

In the second half of 2025, several concurrent works extend "thinking with images" to videos, mostly by improving temporal evidence seeking. VITAL (Zhang et al., 2026) enables an agent to crop temporally relevant clips on demand via a visual toolbox, while LongVT (Yang et al., 2026) and VideoZoomer (Ding et al., 2026) iteratively invoke temporal zoom-in tools to retrieve relevant clips. Conan (Ouyang et al., 2026) teaches the model to identify evidence frames, perform cross-frame deduction, and decide whether to conclude or continue exploring the video. VTimeCoT (Zhang et al., 2025b) proposes a training-free "thinking by drawing" scheme that uses progress-bar tools to improve temporal grounding and reasoning. STVG-o1 (Gu et al., 2025) instead targets spatio-temporal video grounding, but is not designed as an end-to-end video QA method. Compared with these concurrent works, our method jointly integrates temporal localization, spatial grounding, and video reasoning within a single model, and performs single-round inference without external tool use, enabling native "thinking with frames" capability.

## A.3. More Details and Ablation on Training Data

**Data Preparation.** Beyond reporting corpus sizes, we describe here the sampling and filtering strategy applied to each source. For temporal grounding data, we adopt strict constraints to ensure annotation quality and to keep the reasoning process manageable. Specifically, for TVG-Coldstart, we retain only samples with chain-of-thought length under 6,000 characters and with ground-truth spans covering less than 70% of the total video duration. The same filtering is applied to Time-R1, resulting in 2.3k samples. For additional temporal grounding video sources (ActivityNet (Caba Heilbron et al., 2015), COIN (Tang et al., 2019), QueryD (Oncescu et al., 2021), QVHighlight (Lei et al., 2021), DiDeMo (Anne Hendricks et al., 2017)), we keep videos of duration between 10 seconds and 3 minutes, further discarding those where the annotated action lasts more than 50% of the video; TVG-RL is filtered with the same rules, and 2.9k samples are randomly selected. For spatial grounding data, we randomly sample 5k instances from both TreeVGR-SFT and VisCoT. For general video QA data, 15k Video-R1 samples are randomly drawn without additional filtering. For PLM-based video dense captioning data (PLM-Rdcap), we initially sample 3k videos for annotation, from which 2k remain after filtering for quality and consistency. This careful selection yields a high-quality dataset that balances temporal, spatial, and general reasoning

*Table 7.* Impact of different amounts of general VideoQA data. 15k achieves the best balance between grounding and general QA performance.

| VideoQA Data | V-STAR (mAM) | VideoMME (Acc) |
|---|---|---|
| w/o Video-R1 data | 33.4 | 60.7 |
| +5k | 33.0 | 63.2 |
| +15k | **33.7** | **63.6** |
| +30k | 31.7 | **63.6** |

tasks. The resulting dataset provides diverse yet clean supervision signals, making it particularly suitable for training and evaluating spatio-temporal reasoning models.

**Ablation on Different Ratios of General VideoQA Data.** To enhance the model's grounding ability, we emphasize temporal and spatial grounding data during training. However, excessive focus on grounding may weaken the model's original strength in general VideoQA. Thus, an important design choice is how much general VideoQA data to include in the STGR dataset. We compare different ratios and evaluate performance on both grounding-oriented (V-STAR) and QA-oriented (VideoMME) benchmarks. As shown in Table 7, adding 15k VideoQA samples from Video-R1 (Feng et al., 2025) significantly improves QA accuracy without harming grounding performance. In contrast, adding 30k yields no further QA gain while slightly reducing grounding accuracy. Therefore, we adopt *15k VideoQA samples as a balanced choice*, offering strong QA capability while preserving grounding ability and maintaining training efficiency.

### A.4. Prompt for Data Annotation

To obtain high-quality spatio-temporal annotations, we design structured prompts for the Gemini 2.5 Pro API, separately tailored to the two data sources described in Section 3: PLM-Rdcap data and temporal grounding datasets. The goal of these prompts is to guide the model to produce question-answer pairs, key frame selection, bounding boxes, and reasoning chains in a consistent JSON format.

For PLM-Rdcap, as shown in Figure 5, the input is the dense video captions and total frame count, and the output is a JSON with *question*, *answer*, *key_frames*, and *reasoning_process*. Since only frame indices are given, we post-process them into timestamps and align reasoning mentions with annotated object names and boxes.

For temporal grounding datasets, as shown in Figure 6, the input includes the annotated segment, video duration, and segment descriptions, and the output JSON contains the *question*, *answer*, *key_frames* with timestamps, objects and boxes, and the spatio-temporal grounded *reasoning_process*.

We further apply strict filtering and consistency checks, retaining only annotations with validated boxes, aligned timestamps, and coherent reasoning. This ensures a high-quality dataset with reliable spatiotemporal evidence, which is essential for robust training and evaluation.

### A.5. More Ablation Studies on Hyper-parameters and Training Objectives

**Adaptive temporal proximity parameter.** As described in the main paper, we adopt an adaptive temporal proximity schedule in the temporal grounding reward, where the parameter $\sigma$ is gradually annealed from a larger value to a smaller one during training. For a single instance, the temporal reward for point supervision can be written as

$$r_t = \exp\left(-\frac{x^2}{2\sigma^2}\right), \tag{6}$$

where $x$ denotes the absolute difference between the predicted timestamp and the ground-truth timestamp. When $\sigma$ is small (e.g., $\sigma = 1$), the reward rapidly decays for moderate temporal errors, leading to sparse rewards during early training stages. In contrast, a large $\sigma$ (e.g., $\sigma = 4$) assigns relatively high rewards to imprecise predictions, weakening the learning signal for fine-grained temporal refinement. The adaptive schedule alleviates both issues by allowing the model to transition from coarse temporal alignment to precise localization. Empirically, both fixed $\sigma = 1$ and $\sigma = 4$ underperform the adaptive schedule on V-STAR, validating the effectiveness of the proposed design.

**Comparison with a non-o3-like variant.** We further study the importance of o3-style spatio-temporal grounded reasoning

*Table 8.* Ablation on the adaptive temporal proximity parameter $\sigma$ on V-STAR.

| Setting | mAM | mLGM |
|---|---|---|
| Open-o3-Video (adaptive $\sigma$) | **33.7** | **46.6** |
| Fixed $\sigma = 1.0$ | 32.6 | 44.5 |
| Fixed $\sigma = 4.0$ | 33.0 | 45.2 |

*Table 9.* Performance comparison between Open-o3-Video and a non-o3-like variant (pure textual reasoning).

| Model | V-STAR (what) | V-STAR (mLGM) | VideoMME | VideoMME (long) | WorldSense | WorldSense (Rec.) |
|---|---|---|---|---|---|---|
| Open-o3-Video | **61.0** | **46.6** | **63.6** | **54.9** | **37.5** | **36.8** |
| Non-o3-like | 58.6 | 41.0 | 62.3 | 52.8 | 37.1 | 36.4 |

by comparing Open-o3-Video with a non-o3-like variant. The non-o3-like model removes spatio-temporal evidence from SFT and does not receive thinking rewards during reinforcement learning, resulting in a text-only reasoning model. As shown in Table 9, removing o3-style grounding consistently degrades performance across multiple benchmarks, including V-STAR, VideoMME, and WorldSense. The degradation is particularly evident on grounding-intensive settings, such as V-STAR mLGM and long video reasoning in VideoMME, indicating that incorporating spatio-temporal supervision and grounding-aware rewards during training plays a critical role in improving grounded video reasoning performance.

### A.6. Results based on Qwen3-VL Models

Beyond the main results on Qwen2.5-VL, we apply the same training data and optimization pipeline to Qwen3-VL models and report the results in Table 10. We observe that Qwen3-VL already exhibits basic spatio-temporal grounding alongside strong high-level understanding. Building on this foundation, our method yields consistent improvements across model scales: mAM/mLGM increase by **+2.7%/+3.6% on the 4B model, +7.5%/+14.6% on the 8B model, and +4.0%/+8.1% on the 32B model**. These results demonstrate that Open-o3-Video can strengthen spatio-temporal grounded reasoning on top of increasingly strong MLLMs.

### A.7. Results on More Benchmarks

As shown in Table 11, Open-o3-Video performs better than the base model on both STAR and CameraBench. On STAR, it improves accuracy by 3.2%, demonstrating that Open-o3-Video can better handle situated reasoning tasks involving spatio-temporal cues. We also evaluate our model on the CameraBench VQA task and compare it with the baseline model as well as models trained without adaptive temporal proximity or without temporal gating. We find that our model performs better than the baseline and shows gains in challenging motion settings, such as confusable motion, motion and steadiness, and different motion speeds. It also outperforms the variants without Ada. or without Gat. These results indicate that both the model and the training techniques remain stable under camera motions that differ from the training data distribution.

We further evaluate grounding generalization on Charades-STA (Sigurdsson et al., 2016) and HCSTVG-v2 (Tang et al., 2021), as shown in Tables 12 and 13. Charades-STA focuses on temporal grounding, where Open-o3-Video approaches specialized methods such as TimeSuite (Zeng et al., 2025) and TRACE (Guo et al., 2025) in the zero-shot setting. HCSTVG-v2 evaluates spatio-temporal grounding, where Open-o3-Video substantially improves over Qwen2.5-VL-7B and shows strong zero-shot transfer beyond the main QA benchmarks.

### A.8. Inference Frame Rate and Efficiency Analysis

We analyze the effect of inference frame rate on long-video understanding using the LongVideo-Reason-eval benchmark, as shown in Table 14. For the comparison between high and low frame rates, we find that higher frame rates (64 frames) give some improvement, but even with only 16 frames, our model performs well and surpasses the baseline and other reasoning models. And increasing the number of frames from 16 to 64 leads to only marginal gains (e.g., $69.2\% \rightarrow 69.4\%$), indicating diminishing returns from denser frame sampling. For variable frame rates, we follow AKS (Tang et al., 2025) and use the key-frame selection strategy. This strategy achieves 70.1% accuracy, demonstrating that key-frame sampling offers a small improvement over uniform sampling when spatio-temporal reasoning is involved.

*Table 10.* Results on V-STAR based on Qwen3-VL models.

| Model | What | When (Temporal IoU) | | Where (Visual IoU) | | Overall | |
| --- | --- | --- | --- | --- | --- | --- | --- |
| | Acc | Chain1 | Chain2 | Chain1 | Chain2 | mAM | mLGM |
| Qwen3-VL-4B | 59.6 | 18.6 | 19.2 | 26.8 | 7.7 | 31.9 | 43.7 |
| Open-o3-Video-4B | **59.7** | **25.1** | **23.8** | **31.2** | **8.1** | **34.6** | **47.3** |
| Qwen3-VL-8B | 42.9 | 23.5 | **24.4** | 27.3 | 7.1 | 28.0 | 34.4 |
| Open-o3-Video-8B | **61.1** | **24.9** | 23.5 | **33.2** | **8.8** | **35.5** | **49.0** |
| Qwen3-VL-32B | 47.5 | 24.5 | 25.7 | 30.9 | 7.5 | 33.9 | 45.6 |
| Open-o3-Video-32B | **64.6** | **26.4** | **26.0** | **33.6** | **11.9** | **37.9** | **53.7** |

*Table 11.* Performance on STAR and CameraBench.

| Models | STAR | CameraBench VQA | | | |
| --- | --- | --- | --- | --- | --- |
| | Overall | Overall | Confusable Motion | Motion and Steadiness | Motion Speed |
| Qwen2.5-VL-7B | 67.3 | 57.5 | 49.3 | 56.7 | 69.0 |
| Open-o3-Video-7B | **70.5** | **58.8** | **51.3** | **57.6** | **69.3** |
| w/o Ada. | 70.1 | 58.5 | 50.0 | 56.7 | 68.7 |
| w/o Gat. | 69.6 | 57.8 | 50.3 | 55.9 | 67.0 |

We also compare inference efficiency on a 50-sample VideoMME subset using vLLM, and 16 uniformly sampled frames, as shown in Table 15. Direct-answer Qwen2.5-VL is faster because it does not generate reasoning chains. Among single-pass reasoning methods, Open-o3-Video uses fewer output tokens and lower latency than Video-R1 and VideoRFT, while avoiding multi-round tool calls or external model interactions.

### A.9. Details of GSPO Training

For completeness, we provide the full formulation of Group Sequence Policy Optimization (GSPO) (Zheng et al., 2025), which is used in our reinforcement learning stage.

Given a query $x$, the model generates a group of $G$ candidate responses $\{y_i\}_{i=1}^{G}$ sampled from the old policy $\pi_{\theta_{\text{old}}}(\cdot|x)$. Each response is scored by a reward function $r(x, y_i)$, and its normalized advantage is computed as

$$\hat{A}_i = \frac{r(x, y_i) - \text{mean}(\{r(x, y_j)\}_{j=1}^{G})}{\text{std}(\{r(x, y_j)\}_{j=1}^{G})}. \tag{7}$$

The importance ratio is defined at the sequence level as

$$s_i(\theta) = \left( \frac{\pi_\theta(y_i|x)}{\pi_{\theta_{\text{old}}}(y_i|x)} \right)^{\frac{1}{|y_i|}} = \exp\left( \frac{1}{|y_i|} \sum_{t=1}^{|y_i|} \log \frac{\pi_\theta(y_{i,t}|x, y_{i,<t})}{\pi_{\theta_{\text{old}}}(y_{i,t}|x, y_{i,<t})} \right), \tag{8}$$

where $|y_i|$ denotes the response length.

The GSPO objective is then

$$J_{\text{GSPO}}(\theta) = \mathbb{E}_{x, \{y_i\} \sim \pi_{\theta_{\text{old}}}} \left[ \frac{1}{G} \sum_{i=1}^{G} \min\left( s_i(\theta)\hat{A}_i, \text{ clip}(s_i(\theta), 1 - \epsilon, 1 + \epsilon)\hat{A}_i \right) \right], \tag{9}$$

with $\epsilon$ controlling the clipping range.

Unlike GRPO, which clips per-token updates, GSPO clips entire responses, thereby aligning reward assignment with optimization granularity. In practice, this leads to more stable gradients and better performance on long chain-of-thought reasoning tasks.

*Table 12.* Zero-shot results on Charades-STA for temporal grounding. Open-o3-Video approaches temporal grounding expert models without using Charades-STA training data.

| Model | R1@0.3 | R1@0.5 | R1@0.7 |
|---|---|---|---|
| *Temporal grounding expert models* | | | |
| TimeSuite | 69.9 | 48.7 | 24.0 |
| TRACE | 58.6 | 40.3 | 19.4 |
| *Zero-shot general video reasoning model* | | | |
| Open-o3-Video-7B | 64.7 | 46.0 | 21.3 |

*Table 13.* Zero-shot results on HCSTVG-v2 for spatio-temporal grounding. m_tIoU measures temporal localization, while m_vIoU and vIoU@k measure spatial-temporal localization quality. Open-o3-Video substantially improves over Qwen2.5-VL-7B.

| Model | m_tIoU | m_vIoU | vIoU@0.3 | vIoU@0.5 |
|---|---|---|---|---|
| Qwen2.5-VL-7B | 22.9 | 13.0 | 15.6 | 6.4 |
| Open-o3-Video-7B | 36.1 | 23.7 | 35.2 | 12.4 |

## A.10. More Details about Test Time Scaling

To further enhance robustness at inference, we adopt a **confidence-aware test-time scaling** procedure, as shown in Figure 4. Given a video question, the model first generates $N$ independent responses in parallel (In our experiments, $N = 8$, with temperature set to 1.0). Each response contains spatio-temporal grounding annotations in the format `<obj>...</obj><box>...</box>`at`<t>...</t>`s, from which we extract the predicted bounding boxes. The corresponding regions are then cropped from the original video frames and paired with the question to form a new input. This input is passed back into the model to obtain a confidence score $s \in \{0, 1, 2\}$, where:

- $s = 2$: the cropped evidence is highly supportive for answering the question,
- $s = 1$: the evidence may be partially useful,
- $s = 0$: the evidence is irrelevant.

Each initial response is assigned a confidence-weighted score by averaging its evidence scores across all mentioned objects. The final prediction is selected via weighted voting over the $N$ responses. This process effectively filters out hallucinated reasoning traces and highlights consistent evidence across responses.

As reported in Table 16, confidence-aware voting consistently improves over *naive majority voting*, achieving +1.0 on WorldSense and +1.0 on VideoMMMU. This demonstrates that our o3-like spatio-temporal evidence not only enhances grounding, but also provides a natural mechanism for scalable inference and self-correction at test time.

## A.11. More Visualizations

As shown in Figure 7,8,9,10, we provide additional qualitative examples to illustrate the spatio-temporal reasoning ability of Open-o3-Video. These visualizations demonstrate that our model can obtain spatio-temporal evidence and achieve better results.

## A.12. More Limitations and Future Works

While our framework demonstrates strong performance, several limitations remain. First, handling longer videos with complex scenes and smaller objects remains challenging, as high-quality spatiotemporal data for such cases remains relatively scarce. Second, reasoning-intensive queries that require multi-step inference beyond direct grounding remain difficult to fully address. Finally, our current design does not integrate audio or speech information, which often carries crucial cues for understanding video content. Future work will extend the approach to longer and more complex videos, enrich supervision for fine-grained object grounding, and broaden the evidence space to speech or ASR transcripts, scene-event graphs, world knowledge, and motion trajectories.

*Table 14.* Ablation on inference frame rate on LongVideo-Reason-eval.

| Models | Qwen2.5-VL | Video-R1 | | VideoRFT | | Open-o3-Video | | |
|---|---|---|---|---|---|---|---|---|
| Number of Frames | 64 | 64 | 16 | 64 | 16 | 64 | 64 (+AKS) | 16 |
| LongVideo-Reason-eval | 59.3 | 68.9 | 67.3 | 69.4 | 68.0 | 69.4 | 70.1 | 69.2 |

*Table 15.* Inference efficiency comparison on a VideoMME subset under unified hardware and inference settings.

| Model | Thinking | Avg. output tokens | Avg. latency (s) |
|---|---|---|---|
| Qwen2.5-VL-7B | no | **2.0** | **0.978** |
| Video-R1-7B | single-pass CoT | 241.2 | 7.026 |
| VideoRFT-7B | single-pass CoT | 250.5 | 7.556 |
| Open-o3-Video-7B | single-pass CoT | 146.1 | 4.530 |

*Table 16.* Test-time scaling results on WorldSense and VideoMMMU, showing that the confidence-aware voting (N=8) with grounded evidence consistently outperforms base model (N=1) and naive majority voting (N=8).

| Setting | WorldSense | VideoMMMU |
|---|---|---|
| Base | 37.5 | 52.3 |
| Majority Voting | 37.3 | 53.1 |
| Confidence-aware Voting | **38.5** | **54.1** |

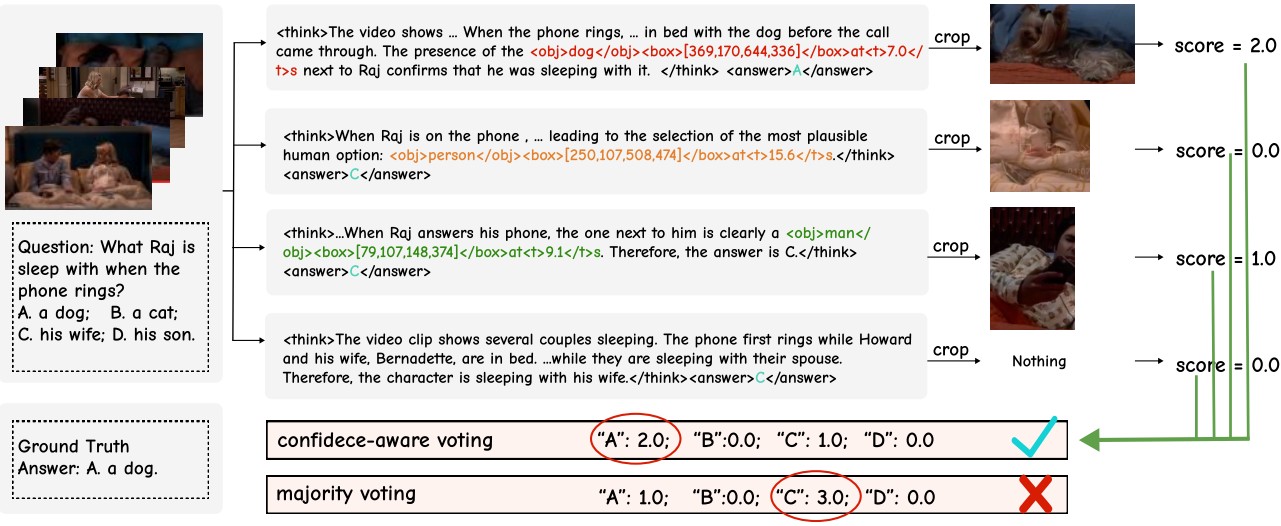

*Figure 4.* Illustration of our **confidence-aware test-time scaling**. The model generates multiple responses with spatio-temporal traces, from which visual regions are cropped and scored for evidence relevance ($s \in \{0, 1, 2\}$). Final predictions are obtained by confidence-weighted voting. Unlike naive majority voting, which is misled by spurious patterns (predicting "C"), our method highlights consistent supportive evidence and correctly predicts "A", thereby improving robustness in inference.

**Prompt for Gemini 2.5 Pro (PLM-Rdcap)**

The video contains a total of {item['total_frames']} frames, with the following dense captions information:
{str(dcap)}
Please complete the following tasks based on the video and caption information:
1. Generate a question-answer pair. Since the dense caption is centered on a specific object or person, the question should also focus on this object or person. You can consider aspects such as its color, clothing, actions, and so on.
2. Output key_frames, which should be the critical frames needed to answer the question. The key_frames must be a list of integer values and fall within the frame range mentioned in the dense caption. (at least one and at most five).
3. Generate a reasoning process:
  - Reasoning must use visual evidence grounded in the video.
  - When referencing the target object or person, you MUST use the following strict format: <obj>object_name</obj>at<t>Frame frame_num</t>
  - The reasoning must not exceed 200 words.
  - The frame number must be in key_frames. The mentioned frame numbers and the visual content of those frames must match consistently.
  - All object names must be identical.
  - Every time you mention the object name (<obj>), you must use the format `<obj>object_name</obj>at<t>Frame frame_num</t>' to specify the corresponding frame.
  - In the reasoning process, except for the text between <t> </t>, the words "frames", "frame" and similar terms MUST not appear.

You must strictly follow the following JSON format (with no additional text outside the JSON):
{{
    "question": "…",
    "answer": "…",
    "key_frames": […],
    "reasoning_process": "…"
}}

*Figure 5.* Annotation Prompt for PLM-Video-Human Region Dense Temporal Captioning Data source.

Prompt for Gemini 2.5 Pro (Temporal Grounding)

The video has a duration of {item['duration']} seconds. The temporal grounding annotation for the video is as follows:
Description: {item['conversations']}
Annotated time segment: {str(item['gt_segment'])}

Based on this annotation, please complete the following tasks:

1. Construct a question-answer pair in open-ended Q&A format.
   - The question should be adapted from the temporal grounding description.
   - The question should focus on a specific object or person, rather than their action.
   - The answer should be concise and not exceed 30 words.
   - Do NOT mention timestamps or annotated time segments in the question.

2. Select at least ONE and at most FIVE keyframes.
   - Each timestamp MUST be within the annotated time segment and be written as a float rounded to exactly one decimal place.

3. For each keyframe, include at least ONE and at most THREE detected objects.
   - Each bounding box coordinates are normalized floats (rounded to exactly two decimal place) in the format [x_min, y_min, x_max, y_max].

4. Generate the reasoning process for answering the question:
   - Reasoning must use visual evidence grounded in the video.
   - When referencing any object, person, or visual element, you MUST use the following strict format:
     <obj>object_name</obj><box>[x_min, y_min, x_max, y_max]</box>at<t>time_in_seconds</t>s
   - Both the `time_in_seconds` and the box coordinates MUST be consistent with the info in the key frames.
   - The reasoning must not exceed 200 words.

You must strictly follow the following JSON format (with no additional text outside the JSON):

```
{{
    "question": "...",
    "answer": "...",
    "key_frames": [
        {{
            "timestamp": time in second,
            "items": {{
                "object1_name": box1,
                "object2_name": box2,
            }}
        }},
        {{
            "timestamp": time in second,
            ...
        }}
    ],
    "reasoning_process": "..."
}}
```

*Figure 6.* Annotation Prompt for Temporal Grounding Data Source.

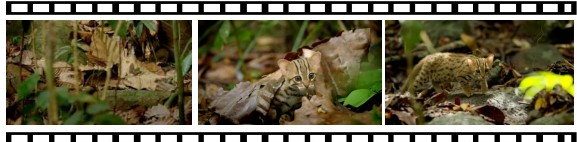

Question: What is the color pattern of the cat in the video?
A. Black and white. B. Calico. C. Solid. D. Spotted.

Ground Truth Answer: D

Qwen2.5-VL-7B: D

Video-R1-7B: \<think\>Let's analyze the color pattern of the cat in the video. The cat has a coat that includes various shades of brown and black. The brown patches are large and distinct, covering a significant portion of the body. The black patterns, including stripes and spots, are scattered throughout the brown areas. The cat does not appear to have any solid color patches, nor does it have a calico pattern, which typically includes orange, black, and white. The color scheme is intricate and multifaceted, which leans more towards a spotted pattern. After considering the detailed analysis, the best fit is the spotted pattern. The intricate mix of brown and black, with spots and stripes, is a hallmark of the "spotted" category. Therefore, the best answer is D.\</think\>
\<answer\>D\</answer\>

VideoRFT-7B: \<think\>To determine the cat's color pattern, I need to identify any visual clues in the video. The text on the red screen mentions "RUSTY SPOTTED CAT," indicating that the cat's color might be rusty. However, the cat's fur is visible, showing a light brown or yellowish color with dark brown spots, resembling a tabby pattern. Given the options, D. Spotted is the closest match, as the spots on the cat's fur align with this pattern.\</think\>
\<answer\>D\</answer\>

Open-o3 Video: \<think\>The video features a cat with a distinct coat pattern. At the very start, the \<obj\>cat\</obj\>\<box\>[55,26,200,200]\</box\> at\<t\>13.0\</t\>s is visible through foliage, and its fur is clearly spotted with a brown base color and black markings. This pattern is consistently seen throughout the video. For example, the \<obj\>cat\</obj\>\<box\>[110,35,349,200]\</box\> at\<t\>51.0\</t\>s's coat is brown with clear black spots. Later, the \<obj\>cat\</obj\>\<box\>[10,110,349,240]\</box\>at\<t\>86.0\</t\>s walks on a branch, and the spotted pattern is maintained. This visual evidence confirms the cat's spotted coloration.\</think\>
\<answer\>D\</answer\>

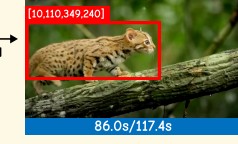

*Figure 7.* **Visualization.** On simple appearance perception tasks, both our model and related baselines can provide correct answers; however, our approach additionally offers explicit spatio-temporal evidence.

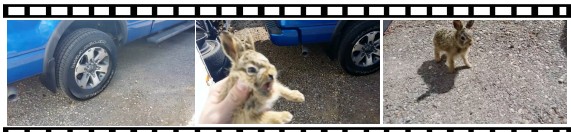

Question: What is the bunny's behavior after being placed on the ground?
A. Finding its mate. B. Attacking the cameraman.
C. Hiding under the car. D. Drilling into the hole.

Ground Truth Answer: B

Video-R1-7B: \<think\>The bunny is initially held by a hand and then placed on the ground. After being placed on the ground, the bunny starts exploring its surroundings. The bunny moves around, sniffing the ground and looking around, which suggests curiosity and exploration. There is no indication that the bunny is looking for its mate or hiding under the car. The bunny is actively moving and exploring. The option of the bunny attacking the cameraman is not evident as the bunny does not show any aggressive behavior towards the camera. The option of the bunny drilling into a hole is also not present in the video, as the bunny is clearly moving around on the ground and not interacting with any holes.
Based on the visual evidence, the bunny's behavior after being placed on the ground is best described as exploring and moving around, which aligns with the option of hiding under the car or finding its mate. \</think\>\<answer\>C\</answer\>

Qwen2.5-VL-7B: B

VideoRFT-7B: \<think\>To determine the bunny's behavior, let's analyze the sequence of events in the video. Initially, the video shows a blue truck and a white car on a gravel surface. A person then picks up a small, light brown bunny from the gravel. The bunny is placed on the ground, and the video captures its behavior after release. The bunny is seen walking away from the car and moving towards the gravel. It then stands on its hind legs, with its front paws lifted, and walks towards the camera, pausing and sniffing the ground. Therefore, the answer is B.\</think\>\<answer\>B\</answer\>

Open-o3 Video: \<think\>The video shows a small, furry bunny. Initially, a hand holds the bunny, and then it is placed on the ground. Immediately, the \<obj\>bunny\</obj\>\<box\>[110,198,302,337]\</box\>at\<t\>18.0\</t\>s is seen on the gravel. The bunny then starts to walk, moving towards the camera. As it walks, the \<obj\>bunny\</obj\>\<box\>[294,34,486,340]\</box\>at\<t\>24.0\</t\>s is observed running on the ground, and its legs are clearly visible. The bunny's natural behavior is to hop and may attack the cameraman\</think\>
\<answer\>B\</answer\>

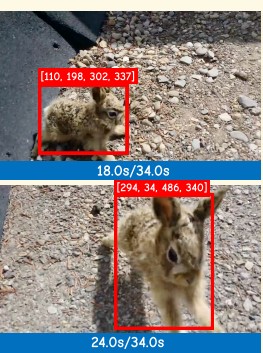

*Figure 8.* **Visualization.** For action recognition, our model precisely localizes both the time and location of the action, achieving superior performance compared to Video-R1.

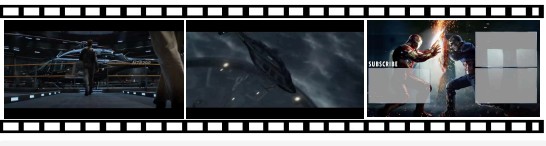

Question: What is the weather like when the helicopter takes off?
A. Snowy. B. Sunny. C. Rainy. D. Cloudless.

Ground Truth Answer: C

Video-R1-7B: <think>As an AI reading the provided video description, it is difficult to tell the weather during the helicopter takeoff accurately due to the absence of visual evidence of weather phenomena such as rain, snow, sunlight, or shadow. The scene involves a black helicopter flying over a body of water that looks dark, reflecting light, which might suggest certain weather conditions. Using the genre of the identified overview video, we can posit that bystanders might perceive the weather as part of the dramatic effect, perhaps with scattered clouds or a suggestive atmosphere. </think>
<answer>A</answer>

Qwen2.5-VL-7B: C

VideoRFT-7B: <think>To determine the weather during takeoff, I first recall the scene. The video initially shows a character in a metallic suit on a platform with a helicopter labeled 'INDUSTRIES' in the background. The scene transitions to an exterior view of the helicopter as it takes off, indicating it is cloudy, with dark blue skies and patches of light breaking through. The helicopter exits the hangar, and the sky remains cloudy throughout the takeoff. There are no visible signs of rain, such as water droplets on the helicopter's surface or a wet ground.
The visual evidence clearly shows the helicopter taking off in a cloudy sky, with no significant presence of rain or snow. The sky is cloudy, and the takeoff sequence is smooth, indicating a cloudy sky.</think>
<answer>A</answer>

Open-o3 Video: <think>The video shows a <obj>helicopter</obj><box>[76,76,353,186]</box>at<t>20.0</t>s taking off from a helipad. There is visible rain or water droplets on the helicopter's surface and the surrounding area, indicating it is raining. The rain continues as the helicopter gains altitude, with the dark, cloudy sky in the background. Therefore, the weather is rainy.</think>
<answer>C</answer>

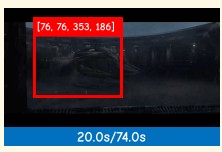

*Figure 9.* **Visualization.** In weather reasoning tasks, our model identifies more effective supporting evidence, whereas related video reasoning models perform poorly.

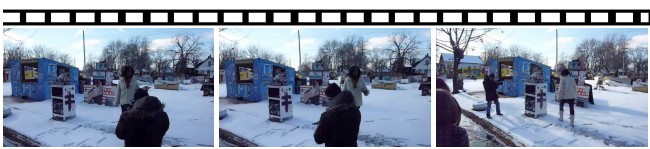

Question: Who uses the camera?

Qwen2.5-VL-7B: The man in the black jacket is using the camera.

Video-R1-7B: <think>In this video, there is a scene where a woman is standing near a snow-covered area with various objects around her. She appears to be... it seems logical that the person holding the camera is the one using it. This person is actively manipulating the camera, adjusting its position, and possibly taking photos or videos of the woman. Therefore, the person holding the camera is most likely the one using it. <answer>The man uses the camera.</answer>

VideoRFT-7B: <think>To determine who is using the camera, let's review the video content step-by-step. Initially, the video shows an outdoor winter scene... However, the video does not explicitly show anyone holding a camera. ...the answer is that no one is using a camera.</think><answer>The person in the light-colored jacket uses the camera.</answer></answer>

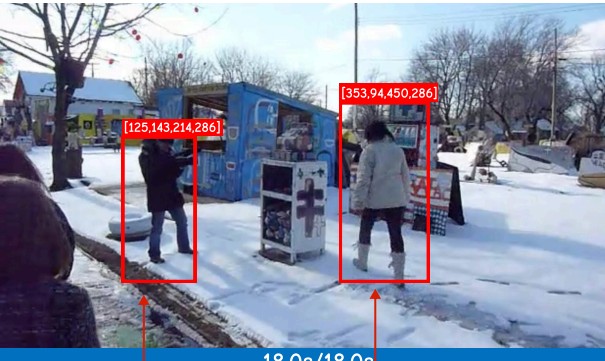

Open-o3 Video: <think>The video shows a woman standing on a snowy street, speaking to the camera. A man is seen holding a camera and filming her. This is evident as the <obj>man</obj><box>[125,143,214,286]</box>at<t>18.0</t>s is positioned to the left of the woman and is actively filming her. The <obj>woman</obj><box>[353,94,450,286]</box>at<t>18.0</t>s is the subject of his filming, as she is the one being recorded. Therefore, it can be concluded that the man is using the camera.</think>\n<answer>A person is using the camera to film the woman standing on the snowy street.</answer>

*Figure 10.* **Visualization.** For open-ended QA, in this example, Video-RFT produces an incorrect analysis, whereas Open-o3-Video answers correctly and provides supporting visual evidence.

