# OpenReview forum: "Open-o3-Video: Grounded Video Reasoning with Explicit Spatio-Temporal Evidence"
_ICML.cc/2026/Conference — ICML 2026 regular_

### Official Review · Reviewer_CVa8 · 2026-02-17

**Soundness:** 3
**Presentation:** 3
**Significance:** 3
**Originality:** 2
**Overall Recommendation:** 5
**Confidence:** 4

**Summary:**

This paper presents Open-o3-Video, a video reasoning model that integrates spatial-temporal grounding during generating Chain-of-Thought (CoT). To train this model, an SFT dataset and an RL dataset containing automatic generated reasoning trajectory data are created. Furthermore, a new reward is designed to stablize the RL training. Open-o3-Video achieves SOTA performance on V-STAR benchmark and shows consistent gains across multiple video understanding benchmarks.  The overall contribution of this work is non-trivial, but the model effectiveness needs stronger justification.

**Compliance With Llm Reviewing Policy:**

Affirmed.

**Final Justification:**

I have no further concerns.

**Key Questions For Authors:**

1. The model is only evaluated on video question answering benchmarks. To demonstrate its reliability in spatiotemporal grounding, the results on Video Temporal Grounding (VTG) benchmarks (e.g., Charades-STA) and Spatio-Temporal Video Grounding (STVG) benchmarks (e.g., HCSTVG and VidSTG) should also be reported.
2. Not all queries require spatio-temporal grounding. Some queries may not involve person or objects. How does the model handle such queries? Can it adaptively determine when to trigger the grounding?
3. Since Open-o3-Video is positioned as a general video reasoning model, it is suggested to include more mainstream video benchmarks, such as MMVU、MVBench and TempCompass. This ensures a comprehensive and consistent comparison against public baselines like Qwen-VL and Video-R1.
4. The data filtering pipeline employs Qwen2.5-VL, which is also the target model used for training. The hallucinations or biases of Qwen2.5-VL might be propagated into the training data. The model may learn to merely mimic spatio-temporal grounding formats without genuinely relying on visual evidence for stronger reasoning.
5. Implementation details:
- What are the inference frame rates used in Tables 1 and 2?
- Is the frame input for Qwen2.5-VL specified as "image" or "video" type? How is the timestamp for each input frame encoded?

**Limitations:**

yes

**Strengths And Weaknesses:**

1. The proposed spatial-temporal grounding CoT is reasonable for improving video reasoning models.
2. While the solution is relatively straightforward, it is non-trivial to sucessfully make it work. I believe this work can provide new insights for future video reasoning works.
3. The paper is well-written.

---

> ### Author Rebuttal · Authors · 2026-03-30
>
> We sincerely thank Reviewer CVa8 for the detailed and constructive feedback. We are encouraged by the reviewer's recognition that this work can provide new insights for future video reasoning works. Below, we address each concern raised by the reviewer.
>
> ***
>
> **Q1: Evaluation on More Grounding Benchmarks**
>
> In the main paper, we have included several grounding benchmarks: V-STAR and TVGBench. Following the reviewer's suggestion, we have supplemented additional evaluations on dedicated grounding benchmarks.
>
> **Charades-STA (Temporal Grounding, Zero-shot)**
>
> |                  | R1@0.3 | R1@0.5 | R1@0.7 |
> | ---------------- | ------ | ------ | ------ |
> | TimeSuite \[1]   | 69.9   | 48.7   | 24.0   |
> | TRACE \[2]       | 58.6   | 40.3   | 19.4   |
> | Open-o3-Video-7B | 64.7   | 46.0   | 21.3   |
>
> On Charades-STA, with <5k out-of-domain temporal grounding training samples in our STGR dataset, our zero-shot performance approaches specialized methods;
>
> **HCSTVG-v2 (Spatio-temporal Grounding, Zero-Shot)**
>
> |                          | m\_tIoU | m\_vIoU | vIoU(0.3) | vIoU(0.5) |
> | ------------------------ | ------- | ------- | --------- | --------- |
> | Qwen2.5-VL-7B (baseline) | 22.9    | 13.0    | 15.6      | 6.4       |
> | Open-o3-Video-7B         | 36.1    | 23.7    | 35.2      | 12.4      |
>
> On HCSTVG-v2, without any dataset-specific fine-tuning, our model improves m\_tIoU and m\_vIoU by over 10% vs. Qwen2.5-VL-7B, demonstrating strong generalization.
>
> ***
>
> **Q2: About Adaptive Grounding**
>
> Our final model does not force spatio-temporal grounding for all queries. Instead, it possesses a degree of adaptive capability to determine whether grounding is necessary.
>
> **For prompting,** we encourage grounding when reasoning about important objects or temporal segments, though not necessarily in every case (we will clarify this in the final version). **For training,** the grounded data naturally demands localization, while general video QA data favors holistic understanding, teaching the model when to ground adaptively. **For inference,** the model produces rich grounding for object/action-dependent tasks (e.g., Figure 8 in the paper), but is more likely to perform textual reasoning directly for semantic analysis questions (example below).
>
> > *Example:*
> >
> > *Q: Which of the following is NOT a reason why there are many teenage mothers in Brazil every year?  A. Abortion is not allowed. B.. C.. D..*
> >
> > *A: \<think>The video provides several reasons for the high number of teenage mothers in Brazil. It mentions that abortion is illegal and dangerous, ..., it does not suggest that ...\</think>\<answer>B\</answer>*
>
> ***
>
> **Q3: Evaluation on More  Video Benchmarks**
>
> | Model                | MVBench | MMVU (mc) |
> | -------------------- | ------- | --------- |
> | Qwen2.5-VL-7B (CoT)  | 59.2    | 60.0      |
> | Video-R1-7B          | 64.8    | 63.8      |
> | VideoRFT-7B          | 62.1    | 68.5      |
> | Open-o3-Video-7B     | 67.6    | 68.0      |
>
> In addition to the benchmarks reported in Table 2, we further evaluate on MVBench and MMVU (multi-choice questions) using 64-frame uniform sampling. The results of other models are taken from their respective papers. As shown above, Open-o3-Video outperforms the Qwen2.5-VL base model and achieves better or comparable performance relative to reasoning models such as Video-R1 and VideoRFT.
>
> ***
>
> **Q4: Potential Bias Propagation**
>
> We acknowledge this potential risk. However, Qwen2.5-VL is only used in a localized verification step, not dominating data generation (Gemini-2.5-Pro is also involved).
>
> Moreover, rather than simply learning the grounding format, our RL thinking reward ensures that a strategy merely mimicking the output without genuinely aligning with visual content receives no effective optimization signals.
>
> Empirically: (1) Table 6 shows that removing model-predicted key evidence frames causes a significantly larger performance drop than random removal; (2) Appendix Table 9 shows that degrading to a non-o3-like variant (no spatio-temporal evidence) leads to clear drops across benchmarks. Both confirm the model genuinely leverages visual evidence rather than superficially imitating the grounding format.
>
> ***
>
> **Q5: Implementation Details**
>
> * Inference frame rates are mentioned in Appendix A.1: 16 frames for V-STAR, 64 frames for others. Frame rate ablation is provided in Appendix A.8.
>
> * Frame input uses the "image" type with timestamps: `Frame {frame_idx} at {time}s: <|vision_start|><|image_pad|><|vision_end|>`.
>
>
>
> We hope the above responses can address the reviewer's concerns.
>
> ***
>
> **References:**
>
> \[1] TimeSuite: Improving MLLMs for Long Video Understanding via Grounded Tuning \[ICLR 2025];
>
> \[2] TRACE: Temporal Grounding Video LLM via Causal Event Modeling \[ICLR 2025].

---

> > ### Author Rebuttal · Reviewer_CVa8 · 2026-04-01
> >
> > Thanks for the authors' reponses in addressing my concerns. I am convinced that Open-o3-Video has learned a reasonable grounding-based reasoning policy. I have no further questions.

---

> > > ### Author Response · Authors · 2026-04-03
> > >
> > > Dear Reviewer CVa8:
> > >
> > > We sincerely appreciate your thoughtful and constructive feedback. We are glad that our rebuttal addresses your concerns, and we appreciate your positive assessment of the paper. We will revise the final version based on your suggestions.

---

### Official Review · Reviewer_AJtp · 2026-03-05

**Soundness:** 3
**Presentation:** 3
**Significance:** 2
**Originality:** 2
**Overall Recommendation:** 4
**Confidence:** 4

**Summary:**

This paper proposes **Open-o3-Video**, a non-agent, single-model framework that makes video MLLMs produce **explicit spatio-temporal evidence** (timestamps + bounding boxes) inside the reasoning trace, aiming for more traceable and verifiable video reasoning.

To enable this, the work (1) curates two training corpora (**STGR-CoT-30k** for SFT and **STGR-RL-36k** for RL) by integrating prior temporal-only/spatial-only resources with 5.9k newly annotated spatio-temporal samples, where each instance includes QA, timestamped keyframes, localized boxes, and a chain-of-thought linking evidence to reasoning steps; and (2) applies a two-stage training recipe (**cold-start SFT + GSPO RL**) with a composite reward including **adaptive temporal proximity** and **temporal gating**, motivated to mitigate a “spatial collapse” issue where spatial rewards are uninformative when temporal localization is still poor.

Empirically, the paper reports large gains on **V-STAR** (Table 1, **+14.4 mAM / +24.2 mLGM** over Qwen2.5-VL) and consistent improvements on multiple video QA / understanding / grounding benchmarks (Table 2).

**Compliance With Llm Reviewing Policy:**

Affirmed.

**Final Justification:**

Most of the concerns have been addressed in the rebuttal, and I have revised my score accordingly.

**Key Questions For Authors:**

Please check Weaknesses (particularly W1-W3.) I will consider revising the score upward if these concerns are addressed or if I have overlooked key evidence/experiments in the current version.

**Limitations:**

yes

**Strengths And Weaknesses:**

## Strengths
1. **Important problem with broad potential impact.** Fine-grained spatio-temporal understanding is a key building block for many downstream applications (e.g., embodied agents / VLA-style systems).
2. **Strong and consistent empirical improvements.** The method improves not only spatio-temporal reasoning on V-STAR, but also general video understanding and temporal grounding benchmarks (Table 2). The paper also includes multiple ablations (training strategy, reward components, and analysis sections).

---

## Weaknesses
1. **Limited algorithmic novelty beyond integration/curation + reward engineering.** Many ingredients—RL post-training for video, temporal grounding RL, spatial grounding supervision, and evidence-centric reasoning formats—are well-aligned with existing research directions. The main contribution reads more like a careful assembly of prior components plus new data + training design.
2. **Reward shaping introduces additional hyperparameters; robustness is unclear.** The proposed “adaptive temporal proximity” and “temporal gating” are the main technical novelty for preventing spatial collapse. However, this design introduces extra hyperparameters (e.g., σ schedule and gating threshold). While the appendix includes an ablation comparing adaptive vs fixed σ (Table 8), it is unclear how robust these choices are and how much tuning would be needed for different datasets/video lengths/noise regimes.
3. **Mandatory grounding at inference may hurt tasks where grounding is unnecessary or ill-posed.** Appendix A.1 states that at test time the model is required to first produce a spatio-temporal grounded reasoning trace, then the final answer. This raises the concern that “always ground it” could induce over-anchoring or spurious evidence selection for more abstract questions or cases where evidence is diffuse.
4. **Benchmark coverage for spatio-temporal grounding could be broader.** V-STAR results are strong, but for a paper emphasizing spatio-temporal reasoning/localization, the evaluation would be more convincing with additional established spatio-temporal grounding benchmarks (beyond a QA-centric benchmark), to test robustness and transfer.
5. **Faithfulness analysis could be more direct.** The paper provides an initial faithfulness check: removing evidence-related frames causes a larger performance drop than randomly removing the same number of frames (Table 6). I would like a more systematic breakdown such as:
   - Answer correct/incorrect × evidence correct/incorrect (e.g., using time error + vIoU thresholds),
   - and quantifying cases where QA is correct but evidence is wrong (or vice versa),
   to assess whether the grounding is truly causal evidence vs post-hoc rationalization.

---

> ### Author Rebuttal · Authors · 2026-03-30
>
> We sincerely thank Reviewer AJtp for the detailed and constructive feedback. We are encouraged by the reviewer's recognition of our work's contributions. Below we address each weakness point by point.
>
> ***
>
> **W1: Algorithmic Novelty**
>
> While SFT+RL training scheme is related to prior work, our core contribution is **realizing explicit spatio-temporal evidence-grounded reasoning**. Unlike methods (Video-R1) that only produce textual thought chains, our model outputs reasoning chains with timestamps and boxes in a single forward pass, enabling test-time scaling via evidence consistency;
>
> To achieve this, we introduce coupled designs: (1) a unified spatio-temporal grounded reasoning dataset; (2) adaptive temporal proximity and gating for stable thinking rewards; and (3) confidence-aware test-time voting. This represents a systematic design across data, training, and inference rather than a trivial combination.
>
> ***
>
> **W2: Reward Design Robustness**
>
> Although our design introduces hyperparameters (*σ* and gating threshold), experiments confirm their robustness. Ablations (Table 8) show fixed *σ* (1 or 4) consistently outperforms the baseline with minimal variance. For temporal gating, we further add ablations with thresholds of 1, 3, and 5s, yielding mLGM scores of 46.6/46.6/46.0 on V-STAR without training instability.;
>
> Furthermore, consistent gains on out-of-distribution benchmarks (course video: VideoMMMU, camera-motion video: CameraBench, long video: LongVideoReason-Eval) demonstrate that our reward design generalizes well across diverse video types, rather than relying on fine-grained parameter tuning.
>
> ***
>
> **W3: About "always ground it"**
>
> Our method does not force grounding on every question. It develops an adaptive evidence expression mechanism.**&#x20;For prompting,** we encourage grounding when reasoning about important objects or temporal segments, though not necessarily in every case (we will clarify this in the final version). **For training,** the grounded data naturally demands localization, while general video QA data favors holistic understanding, teaching the model when to ground adaptively. **For inference,&#x20;**&#x74;he model produces rich grounding for object/action-dependent tasks (e.g., Figure 8 in the paper), but is more likely to perform textual reasoning directly for semantic analysis questions (example below).
>
> > *Example:*
> >
> > *Q: Which of the following is NOT a reason why there are many teenage mothers in Brazil every year?  A. Abortion is not allowed. B.. C.. D..*
> >
> > *A: \<think>The video provides several reasons for the high number of teenage mothers in Brazil. It mentions that abortion is illegal and dangerous, ..., it does not suggest that ...\</think>\<answer>B\</answer>*
>
> ***
>
> **W4: Additional Grounding Benchmarks**
>
> We further evaluate on standard grounding benchmarks beyond V-STAR and TVGBench.
>
> **Charades-STA (Temporal Grounding, Zero-shot)**
>
> |                | R1@0.3 | R1@0.5 | R1@0.7 |
> | -------------- | ------ | ------ | ------ |
> | TimeSuite \[1] | 69.9   | 48.7   | 24.0   |
> | TRACE \[2]     | 58.6   | 40.3   | 19.4   |
> | Open-o3-Video  | 64.7   | 46.0   | 21.3   |
>
> On Charades-STA, with less than 5k out-of-domain temporal grounding training samples in our STGR dataset, our model shows strong zero-shot performance compared with specialized methods;
>
> **HCSTVG-v2 (Spatio-temporal Grounding, Zero-shot)**
>
> |                       | m\_tIoU | m\_vIoU | vIoU@0.3 | vIoU@0.5 |
> | --------------------- | ------- | ------- | -------- | -------- |
> | Qwen2.5-VL (Baseline) | 22.9    | 13.0    | 15.6     | 6.4      |
> | Open-o3-Video         | 36.1    | 23.7    | 35.2     | 12.4     |
>
> On HCSTVG-v2, without any dataset-specific fine-tuning, our model improves m\_tIoU and m\_vIoU by over 10% vs. Qwen2.5-VL baseline, demonstrating strong generalization.
>
> ***
>
> **W5: Evidence-Answer Consistency**
>
> We analyze the relationship between grounding evidence and answer correctness following the settings of V-STAR. (QA,T,S denote QA, temporal and spatial accuracy, respectively.)
>
> | Accuracy (Chain 1) | QA   | S\&T | QA\&S\&T |
> | ------------------ | ---- | ---- | -------- |
> | Qwen2.5-VL         | 33.5 | 11.3 | 4.5      |
> | Open-o3-Video      | 61.0 | 22.3 | 15.3     |
>
> * **When QA is correct**, spatio-temporal evidence is also correct: Qwen2.5-VL: 4.5/33.5 = 13.4%, Open-o3-Video: 15.3/61.0 = **25.1%**.
>
> * **When evidence is correct**, QA is also correct: Qwen2.5-VL: 4.5/11.3 = 39.8%, Open-o3-Video: 15.3/22.3 = **68.6%**.
>
> Although not every correctly answered question is accompanied by precise grounding, these results show that our method achieves much stronger consistency between evidence and answer compared to Qwen2.5-VL.
>
> We hope the above responses can address the reviewer's concerns.
>
> ***
>
> **Reference**
>
> \[1] TimeSuite: Improving MLLMs for Long Video Understanding via Grounded Tuning \[ICLR2025]
>
> \[2] TRACE: Temporal Grounding Video LLM via Causal Event Modeling \[ICLR2025]

---

> > ### Author Rebuttal · Reviewer_AJtp · 2026-04-03
> >
> > Most of the concerns have been addressed in the rebuttal, and I have revised my score accordingly.

---

> > > ### Author Response · Authors · 2026-04-03
> > >
> > > Dear Reviewer AJtp:
> > >
> > > We sincerely appreciate your thoughtful and constructive feedback. We are glad that our rebuttal addresses most of your concerns, and we appreciate your positive assessment of the paper. We will revise the final version based on your suggestions.

---

### Official Review · Reviewer_31nM · 2026-03-12

**Soundness:** 3
**Presentation:** 3
**Significance:** 3
**Originality:** 2
**Overall Recommendation:** 4
**Confidence:** 4

**Summary:**

This paper introduces Open-o3-Video, an innovative agentless video reasoning framework that directly generates reasoning chains with explicit timestamps and bounding boxes through a single forward pass. Coupled with the newly constructed STGR spatio-temporal dataset and an adaptive reinforcement learning strategy, it achieves state-of-the-art performance on key benchmarks such as V-STAR and TVGBench. This significantly enhances the interpretability and accuracy of video reasoning while maintaining strong competitiveness in general question-answering tasks. TVGBench benchmarks, significantly enhancing the interpretability and accuracy of video reasoning while maintaining strong competitiveness in general question-answering tasks.

**Compliance With Llm Reviewing Policy:**

Affirmed.

**Final Justification:**

Most of my questions have been answered, and I still maintain my score

**Key Questions For Authors:**

1. Does the performance drop on WorldSense stem from architectural overhead or data bias? An ablation study without grounding objectives would clarify if this trade-off is inherent, significantly impacting my view on the model's general utility.

2. Please provide concrete inference latency (tokens/sec) and memory usage compared to Qwen2.5-VL. Demonstrating marginal overhead is critical to validate the "single-pass" efficiency claim for real-time deployment.

3. Have you verified thecausal consistency between generated evidence and reasoning (e.g., via human eval) to rule out "coordinate hallucinations"? Confirming true grounding rather than formatted output is essential to support the claimed verifiability.

**Limitations:**

Although the authors indirectly demonstrate the model's performance trade-offs in pure perception tasks through its WorldSense benchmark results in the experimental section, the paper lacks an in-depth discussion of this limitation and an analysis of its underlying causes. Furthermore, the article completely omits any consideration of potential negative societal impacts, such as: explicitly generated spatio-temporal evidence, if maliciously exploited, could enhance the realism of deepfakes or lead to more covert privacy violations (such as automated precision tracking).

**Strengths And Weaknesses:**

**Strengths:**

1.The paper proposes the Open-o3-Video framework, overcoming the limitations of traditional video reasoning models that either output text-based answers or rely on multi-step agent invocations.

2.To address the lack of joint spatio-temporal supervision in existing datasets, the authors constructed the STGR-CoT-30k and STGR-RL-36k datasets, filling a gap in the field.

3.The experimental results are compelling, with the model significantly outperforming baselines (such as Qwen2.5-VL) and closed-source models (GPT-4o) on core tasks including V-STAR (a specialized spatio-temporal reasoning benchmark), TVGBench (a localization benchmark), and LRR (long video reasoning).

**Weaknesses:**

1.As shown in Table 2, the model's performance on the Overall and Recognition metrics (37.5 / 36.8) on the WorldSense dataset falls below that of some comparison models (e.g., InternVL-2.5-8B's 39.6 / 38.5). This indicates that incorporating complex spatio-temporal localization heads and associated training objectives may have partially constrained the model's capacity for pure visual perception or multimodal common-sense reasoning, revealing a performance trade-off.

2.The construction of the STGR dataset relies on cleaning and integrating existing data alongside newly added high-quality manual/semi-automated annotations (5.9k samples). This high-cost annotation dependency requires precise timestamps and bounding boxes, which may limit the method's scalability for future large-scale deployment. Without cost-effective access to additional similar spatio-temporal supervised data, the model's generalization capabilities could be constrained.

3.Although the paper highlights the efficiency of ``single-pass forward propagation,'' generating long sequence outputs containing detailed reasoning chains, multiple timestamps, and bounding boxes inevitably increases token generation volume and inference latency. The study lacks comparative analysis of inference speed (Tokens/s), GPU memory consumption, and efficiency versus traditional baseline models on equivalent hardware, which is a critical omission for practical deployment.

4.Although metrics like mAM and mLGM have been introduced to evaluate spatio-temporal alignment, current assessments primarily rely on automated metrics or simple accuracy rates for evaluating the logical quality of generated chains of thought (CoT) and the causal consistency between evidence and reasoning. There is a lack of deeper human evaluation or finer-grained metrics to measure whether models truly “understand” evidence or merely learn to output coordinates in a ‘formatted’ manner (i.e., whether “coordinate hallucinations” exist where answers happen to be correct).

---

> ### Author Rebuttal · Authors · 2026-03-30
>
> We sincerely thank Reviewer 31nM for the detailed and constructive feedback. We are encouraged by the reviewer's recognition that our Open-o3-Video framework overcomes the limitations of traditional video reasoning models by directly generating reasoning chains with explicit timestamps and bounding boxes through a single forward pass. Below, we address each weakness (W) and question (Q) in detail.
>
> ***
>
> **W1\&Q1: Performance on WorldSense and Ablation on Grounding Objectives**
>
> We thank the reviewer for pointing out this observation. On WorldSense, our method improves over the base model Qwen2.5-VL-7B (36.1 vs. 37.5 in Table 2), indicating that incorporating spatio-temporal grounding does not weaken the original model's understanding capability. We agree that a gap remains compared to certain video foundation models (e.g., InternVL-2.5) on WorldSense , which reflects differences in capability emphasis across models.
>
> Regarding the ablation study, we have provided a comparison in the Appendix A.5 (Table 9): a **non-o3-like variant** is constructed by removing spatio-temporal evidence supervision and associated rewards, degenerating into a pure text reasoning paradigm. This variant shows performance degradation across multiple benchmarks, demonstrating that introducing grounding objectives does not weaken general capabilities under our training paradigm, but rather enhances the model's ability to ***think with videos.***
>
> ***
>
> **W2: Data Cost and Scalability**
>
> We thank the reviewer for this concern. We agree that high-quality spatio-temporal annotations are costly to construct, which is a common challenge in this direction. However, our pipeline does not rely on large-scale human annotation. The 5.9k samples are primarily generated through an automated process (model generation + multi-stage filtering), with limited human involvement. This pipeline can be readily reused on more video data to scale up. Moreover, the substantial performance gains from only 5.9k samples suggest that our method exhibits good data efficiency.
>
> ***
>
> **W3\&Q2: Inference Efficiency and Deployment Cost**
>
> We thank the reviewer for this concern. We conducted an efficiency comparison on a VideoMME subset (50 samples) under unified hardware (single 80GB H100) and inference settings (vLLM, 16 uniformly sampled frames):
>
> |                | Thinking        | Avg output tokens  | Avg latency (s)    |
> | -------------- | --------------- | ------------------ | ------------------ |
> | Qwen2.5-VL-7B  | no              | 2.0                | 0.978              |
> | Video-R1       | single-pass CoT | 241.2              | 7.026              |
> | VideoRFT       | single-pass CoT | 250.5              | 7.556              |
> | Open-o3 Video  | single-pass CoT | **146.1**          | **4.530**          |
>
> *Table: Comparison of output tokens and latency across models.*
>
>
>
> Compared to direct-answer Qwen2.5-VL, reasoning indeed incurs greater overhead. However, among reasoning methods, our approach has lower inference cost than both Video-R1 and VideoRFT, indicating good generation efficiency while incorporating spatio-temporal evidence. Furthermore, our method requires no multi-round tool invocations or external model interactions, leading to lower scheduling overhead and better deployment simplicity at the system level.
>
> ***
>
> **W4\&Q3: Causal Consistency Between Evidence and Reasoning**
>
> We appreciate this insightful question. To address this concern, we provide the following evidence:
>
> (1) As mentioned in Section 3.2 ("Self-consistency Checking and Quality Control"), language descriptions and extracted bounding boxes are required to be semantically consistent, ensuring alignment between reasoning and evidence.
>
> (2) Both Table 6 in Section 5.2 and Table 9 in Appendix A.5 demonstrate that grounded evidence within the chain-of-thought improves reasoning ability, rather than merely producing formatted localization outputs.
>
> We agree with the reviewer's suggestion and plan to incorporate more systematic human evaluation in future work to further validate this point.
>
> ***
>
> **Limitations: Societal Impact**
>
> We acknowledge that spatio-temporal evidence, if maliciously exploited, could pose risks such as enhancing deepfake realism or enabling covert privacy violations. We will add a discussion of potential risks and mitigation strategies in the revised manuscript.
>
> ***
>
> We hope the above responses can address the reviewer's concerns. We will incorporate the discussed revisions into the updated paper.

---

> > ### Author Rebuttal · Reviewer_31nM · 2026-04-02
> >
> > Most of my questions have been answered, and I still maintain my score

---

> > > ### Author Response · Authors · 2026-04-03
> > >
> > > Dear Reviewer 31nM:
> > >
> > > We sincerely appreciate your thoughtful and constructive feedback. We are glad that our rebuttal addresses most of your concerns, and we appreciate your positive assessment of the paper. We will revise the final version based on your suggestions.

---

### Official Review · Reviewer_JHYV · 2026-03-12

**Soundness:** 3
**Presentation:** 3
**Significance:** 3
**Originality:** 3
**Overall Recommendation:** 4
**Confidence:** 3

**Summary:**

This paper introduces Open-o3-Video, a framework for injecting evidence into video reasoning models. The authors start by constructing training datasets: STGR-Cot-30k and STGR-RL-36k, which are existing datasets with augmented evidence annotations. With the dataset, the authors propose adaptive temporal proximity and temporal gating to stabilize RL finetuning for the video reasoning task. Evaluations are conducted on V-STAR, VideoMME, and WorldSense, etc, to demonstrate the efficacy of the model.

**Compliance With Llm Reviewing Policy:**

Affirmed.

**Final Justification:**

As summarized in my acknowledgement, I will maintain my original score.

**Key Questions For Authors:**

See weaknesses.

**Limitations:**

The limitation section should be placed in the main paper, not the appendix.

**Strengths And Weaknesses:**

Strength:
- This seems to be the first work trying to inject evidence-based reasoning into video reasoning tasks. This is well-motivated.
- The two training datasets and the insights in RL finetuning are favorable for the community.
- The results are solid and extensive.

Weaknesses:
1. Why are the improvements much smaller in Table 2 compared to Table 1? I am a little concerned that the model is a bit overfitting to the V-STAR benchmark.
2. The evidence is only about the bounding box, objects, and the time; this might not capture all the cues that are useful in reasoning about a video. I wonder if the system can be further augmented/extended to more types of evidence.
3. Some of the baselines are rather old, e.g., how does Gemini3 perform in V-STAR and the benchmarks in Table 2? The authors should discuss the most SOTA foundation models and better demonstrate their limitations in video reasoning tasks to make the paper stronger.

---

> ### Author Rebuttal · Authors · 2026-03-30
>
> We sincerely thank Reviewer JHYV for the detailed and constructive feedback. We are encouraged by the reviewer's recognition that this seems to be the first work trying to inject evidence-based reasoning into video reasoning tasks and that it is well-motivated. Below, we address each concern point by point.
>
> ***
>
> **W1: Differences in improvement between Table 1 and Table 2**
>
> We appreciate this thoughtful concern. We would like to clarify the rationale behind our benchmark selection and the different focuses of Table 1 and Table 2.
>
> **Regarding the choice of V-STAR:&#x20;**&#x57;e select V-STAR because it comprehensively evaluates QA, temporal grounding, and spatial grounding in a unified manner, which aligns well with our evidence-grounded video reasoning framework. Moreover, our training data does not overlap in distribution with V-STAR, so the improvements are not due to overfitting.
>
> **Regarding the difference between Table 1 and Table 2:&#x20;**&#x54;he two tables have distinct emphases. Table 1 focuses on spatio-temporal grounding, where our method shows larger gains. Table 2, on the other hand, largely emphasizes general video understanding and reasoning. Together, the two tables demonstrate that our method not only enhances fine-grained evidence localization ability but also maintains good general understanding and reasoning performance.
>
> ***
>
> **W2: Evidence Extensibility**
>
> We agree that this is an important direction for future exploration. In the context of combining visual information for reasoning, time, objects, and bounding boxes represent some of the most critical factors, which is why our current framework primarily focuses on these cues.
>
> And we acknowledge that there are many other valuable types of evidence worth incorporating in future work, such as **speech/ASR transcripts, scene-event relational graphs, world knowledge, and motion trajectories**, all of which could meaningfully enrich the reasoning process.
>
> We would like to emphasize that our framework is designed with extensibility and has the capacity to accommodate new types of evidence. However, supporting additional evidence types with high-quality chain-of-thought annotations would require more costly data labeling efforts. In terms of extensibility, for CoT construction, one could design dedicated tags within the reasoning chain, presenting evidence information in a structured data format within each tag. For RL training, corresponding reward signals can be added as new sub-components of the thinking reward.
>
> ***
>
> **W3: Comparison with SOTA models like Gemini 3**
>
> Thank you for this valuable suggestion. We have conducted additional experiments with Gemini-3 Pro on the V-STAR benchmark. The results are as follows:
>
> | Model         | What | Chain-1 (tIoU) | Chain-2 (tIoU) | Chain-1 (vIoU) | Chain-2 (vIoU) | mAM  | mLGM |
> | ------------- | ---- | -------------- | -------------- | -------------- | -------------- | ---- | ---- |
> | Gemini-3 Pro  | 59.1 | 24.2           | 23.8           | 7.2            | 4.8            | 29.9 | 41.2 |
> | Open-o3-Video | 61.0 | 24.5           | 24.0           | 25.4           | 6.0            | 33.7 | 46.6 |
>
> *Table: Performance on V-STAR Benchmark*
>
> Gemini 3 Pro surpasses other closed-source models in Table 1 of our paper but still scores relatively low on dense spatial grounding. Open-o3-Video outperforms Gemini 3 overall, with comparable temporal grounding and a clear advantage in spatial localization (Chain-1 vIoU: 25.4 vs. 7.2), further validating the effectiveness of our evidence-grounded reasoning approach. We will include this comparison in the revised version.
>
> ***
>
> **Limitation section placement**
>
> We appreciate this suggestion and will move the limitation section from the appendix to the main paper in the revised version.
>
> ***
>
> We hope these responses can address the reviewer's concerns. We are committed to incorporating all suggested improvements in the revised version and welcome any further feedback.

---

> > ### Author Rebuttal · Reviewer_JHYV · 2026-04-03
> >
> > Thanks for addressing my concerns, the addition comparison makes the paper stronger, I will maintain my positive score.

---

> > > ### Author Response · Authors · 2026-04-03
> > >
> > > Dear Reviewer JHYV:
> > >
> > > We sincerely appreciate your thoughtful and constructive feedback. We are glad that our rebuttal addresses your concerns, and we appreciate your positive assessment of the paper. We will revise the final version based on your suggestions.

---

### Decision · Program_Chairs · 2026-04-30

**Decision:**

Accept (regular)

**Comment:**

This paper considers the problem of spatio-temporal LLM reasoning, proposing Open-o3-video, which generates reasoning chains for video question answering with timestamps, localized bounding boxes, and a chain of thought explicitly linking visual evidence to reasoning steps. The paper introduces two training sets, STGR-CoT-30k and STGR-RL-36k, for supervised fine-tuning and RL, integrating temporal-only and spatial-only datasets. A key innovation in Open-o3-video is an adaptive temporal proximity approach that uses a coarse-to-fine reward scheme for temporal localization as training progresses; this localization precision is then used in a temporal gating scheme to compute spatial rewards. Experiments on the V-STAR benchmark demonstrate strong results, surpassing GPT-4o and other models.

The paper received overall positive reviews, with three weak accept and one accept recommendation. Reviewers generally appreciated the evidence-based video reasoning approach, the potential usefulness of the two proposed datasets, and the strong empirical results. Concerns were raised regarding the limited notion of “evidence,” focusing mainly on bounding boxes, objects, and timestamps, which may be insufficient for holistic spatio-temporal understanding (JHYV, AJtp, CVa8); slightly outdated baselines or limited experimental comparisons (JHYV, AJtp, CVa8); data cost, scalability, and efficiency (31nM); and limited algorithmic novelty (AJtp). The authors provided a strong rebuttal, including additional SOTA comparisons with Gemini 3 Pro, further analysis of dataset scalability and inference efficiency, and additional results on Charades-STA and HCSTVG-v2.

The AC conducted an independent reading of the paper and agrees with the reviewers that it makes a substantial contribution to spatio-temporal grounded video reasoning using VLMs, and thus recommends acceptance. The authors are encouraged to incorporate the additional experiments presented during rebuttal into the main paper.